# Influence of Selected Hypromellose Functionality-Related Characteristics and Soluble/Insoluble Filler Ratio on Carvedilol Release from Matrix Tablets

**DOI:** 10.3390/pharmaceutics17101358

**Published:** 2025-10-21

**Authors:** Tadej Ojsteršek, Grega Hudovornik, Franc Vrečer

**Affiliations:** 1KRKA, d. d., 8501 Novo Mesto, Slovenia; 2Faculty of Pharmacy, University of Ljubljana, 1000 Ljubljana, Slovenia

**Keywords:** hypromellose, HPMC, controlled release, modified release, extended release, prolonged release, drug release, functionality-related characteristics

## Abstract

**Background/Objectives:** This study investigated how selected functionality-related characteristics (FRCs) of hypromellose (HPMC)—namely viscosity, hydroxypropoxy substitution, particle size, and the ratio of water-soluble (FlowLac^®^ 100) to water-insoluble (Avicel^®^ PH-102) fillers— affect the release of carvedilol from matrix tablets. **Methods:** Using a Central Composite Design (CCD) Design of Experiments (DoE), mixtures of HPMC QbD samples were prepared to achieve target HPMC FRC levels. Within the CCD, levels of FlowLac^®^ 100 and Avicel^®^ PH-102 were also varied. The mean and standard deviation of carvedilol release at each analyzed time point of the release profile were used as target variables for individual multiple linear regression (MLR) models. **Results:** Lactose, the water-soluble filler, significantly accelerated carvedilol release, whereas the water-insoluble MCC slowed and stabilized release by improving gel integrity. Among the HPMC FRCs, particle size had the strongest influence during the early release phase, while HPMC viscosity and hydroxypropoxy substitution degree became more important in later phases. Analysis of the results using optimized multiple linear regression (MLR) models revealed key interaction effects, particularly between HPMC viscosity and lactose content, and between viscosity and particle size, demonstrating their combined role in modulating release kinetics. **Conclusions:** These findings provide valuable insight into how controlling HPMC’s FRCs and filler composition can reduce interbatch variability in drug release and support the rational design of robust controlled release formulations.

## 1. Introduction

Hypromellose (HPMC)-based hydrophilic matrix tablets are one of the most widely used and most common types of oral controlled release (CR) dosage forms for achieving prolonged drug release in the gastrointestinal tract (GIT) [1,2,3,4,5,6,7,8]. They provide all the principal therapeutic benefits of CR oral dosage forms like reduced dosing frequency (improving patient compliance) and more stable plasma drug concentrations, leading to potential better therapeutic outcomes and less unwanted side effects [3,4,8,9]. Moreover, they can be manufactured by simple, traditional methods (direct compression or wet granulation), making them very competitive in cost compared to other CR technologies [1,2,3,4,8]. HPMC has a GRAS (generally recognized as safe) status, is available in several different substitution types and/or viscosity grades, and being a non-ionic polymer, its viscosity (and hence release behaviour) is pH-independent across a broad range (pH 3–11) [1,2,3]. HPMC is a very common, readily available, and relatively inexpensive polymer compared to many other pharmaceutical-grade polymers used for controlled release, which all contribute to the frequent use of this polymer in CR formulations [1,2,3]. For SR tablets, the 2208 substitution type is the one most frequently used, although the 2910 substitution type has also been used as a hydrophilic matrix former [1].

HPMC functions as a hydrophilic matrix former by initially wetting and hydrating upon contact with aqueous media [1,2,10,11,12]. As water penetrates into the tablet, HPMC swells, increasing in volume and forming a gel layer on the tablet’s surface [1,2,3,11,12,13]. This gel is a physical (non-chemically crosslinked) network, held together by polymer chain entanglements, hydrogen bonds, and hydrophobic associations, rather than covalent bonds [14,15,16]. The gel layer serves as a barrier both to further water ingress and to the diffusion of dissolved drugs (and any soluble excipients) into the surrounding medium [8,10]. For poorly water-soluble drugs, matrix erosion often predominates and can become the rate-limiting step in release [1,2,3,10,13,17]. This erosion also counteracts the lengthening of the diffusional path as the gel layer thickens, helping to maintain a more constant release rate [2,12,17,18]. Some authors identify the diffusion front as the principal driver of the overall release kinetics, although all three fronts—the swelling front, the diffusion front, and the erosion front—contribute to drug release [17,19].

HPMC has three primary functionality-related characteristics (FRCs): viscosity, degree of substitution, and particle size (distribution) [1,2,20,21,22,23]. Additional FRCs, such as substitution homogeneity/heterogeneity [24], particle shape [25,26], and other particulate properties, are less commonly cited and, to our knowledge, have not been systematically investigated via design of experiments (DoE) to avoid confounding with the primary FRCs.

Manufacturers provide Quality by Design (QbD) samples with varied viscosity within the same grade and substitution type [21,22,23], enabling studies of HPMC viscosity’s impact on drug release within the batch-to-batch variability framework. Higher-viscosity HPMC forms a stronger and more viscous gel layer upon hydration, reducing erosion in vitro and in vivo, and slowing diffusion of drug, water, and soluble excipients compared to lower-viscosity grades [1,2,3,10,20,21,23,26,27]. Increased viscosity also correlates with reduced surface hydrophilicity and may be influenced by higher MeO substitution [28]. While lower-viscosity HPMC hydrates and swells faster, at least in lower loadings in tablets [29,30], higher-viscosity grades absorb more water into their gel network, significantly limiting erosion compared to their lower-viscosity counterparts [31].

Within the same HPMC viscosity grade and substitution type, HP substitution is more likely than MeO substitution to influence functionality. Manufacturers provide QbD samples with varied HP substitution, but not MeO, within specification ranges [21,22,23], enabling studies of HP substitution’s impact on drug release. MeO substitution has a relatively narrower specification range than HP [32], suggesting less batch-to-batch variation. Studies in published literature primarily focus on HP substitution’s effect on drug release, with limited research on MeO substitution within the same HPMC viscosity grade and substitution type [1,22,33,34].

The degree of MeO and HP substitution influences HPMC hydrophilicity. Substitution of hydrophilic hydroxyl groups generally reduces overall hydrophilicity of HPMC, but the specific substituent type (MeO or HP) affects hydration, swelling, and water/solute transport. MeO groups are generally more hydrophobic than HP and reduce HPMC swelling to a greater extent [1]. HP substitution is a significant factor influencing drug release, particularly for poorly soluble drugs where polymer erosion is the predominant release mechanism [1,2]. This influence is more evident with lower HPMC tablet content [26]. Higher HP substitution increases hydrophilicity, resulting in faster hydration and swelling [35]. Furthermore, higher HP substitution correlates with enhanced polymer relaxation and erosion, thereby increasing the contribution of the erosion release mechanism, whereas lower HP substitution favours the diffusional release mechanism [36].

Some researchers also focused on the homogeneity/heterogeneity of HPMC substitution and concluded that this could also be a factor in drug release [23,24,37]. QbD samples obtained from HPMC producers with different homogeneity/heterogeneity of substitution, however, have not yet been reported in any studies.

Manufacturers supply HPMC QbD samples with defined fine and coarse particle fractions [21,22,23], which can be used to study how HPMC particle size impacts drug release. Multiple studies demonstrate that finer HPMC particles hydrate and swell more rapidly, forming a uniform gel barrier sooner, and thus more effectively retarding drug release than coarser particles. Fine particles also enhance polymer–drug/excipient interactions and dispersibility, whereas coarse particles create larger interparticulate pores that accelerate water ingress, increase initial burst release (an initial uncontrolled release of the drug from and near the surface of the tablet), and, in extreme cases, cause matrix failure. These particle-size effects are most pronounced at low HPMC loadings—where coarse particles particularly exacerbate burst release—and diminish at higher loadings. Furthermore, beyond a critical reduction in particle size, no additional control over drug release is observed [1,2,26]. Although other particulate attributes may influence drug release, they remain largely unexplored, and HPMC QbD samples are currently differentiated solely by mean particle size [21,22,23].

Interbatch variability in HPMC’s primary FRCs—viscosity, HP substitution, and particle size—can compromise the reproducibility of dissolution profiles for HPMC-based SR tablets [1,22,23,33,34,38]. Since identical FRC levels are unattainable from manufacturers, compensatory strategies are needed, potentially via variable composition and statistical modelling. Inclusion of water-soluble fillers in HPMC matrix tablets results in faster drug release, whereas inclusion of water-insoluble fillers in slower drug release [1,2,39,40,41,42,43,44,45,46,47,48,49,50,51,52,53]. Fillers also influence burst release, lag time (delayed onset of drug release), and intrabatch drug release variability [53]. Here, we evaluate whether adjusting levels of lactose monohydrate (FlowLac^®^ 100) and microcrystalline cellulose (Avicel^®^ PH-102) can offset interbatch FRC variability in carvedilol (model poorly water-soluble drug) SR tablets based on HPMC 2208 K15M. FlowLac^®^ 100 and Avicel^®^ PH-102 were selected as water-soluble and water-insoluble modifiers of carvedilol release, respectively, based on their previous performance regarding mean carvedilol release, intrabatch release variability, burst effect, and direct compression processability [53].

No study to date has applied a high-resolution DoE [54,55,56,57,58] to simultaneously vary and control the primary HPMC FRCs—viscosity, HP substitution, and particle size. Available HPMC QbD samples [21,22,23] vary one FRC at a time without adjusting the levels of the others according to a full factorial or response-surface (RSM) design. Consequently, only main HPMC FRC effects have been studied, with potential second-order interactions—likely relevant to drug release—left unexplored. Similarly, second-order interactions between individual HPMC FRCs and the proportion of each filler in the water-soluble/water-insoluble filler mixture within the tablet were also not explored. This study builds upon a previous study by Košir et al. [22], which also deals with the influence of HPMC FRCs on carvedilol release, but is performed using a different type of HPMC (HPMC 2208 K15M instead of HPMC 2208 K4M), enhanced with a higher-resolution DoE, and extended with a study into filler’s (water-soluble vs. water-insoluble) impact on carvedilol release.

## 2. Materials and Methods

### 2.1. Materials

Carvedilol (free base) was supplied by Krka, d.d., Novo Mesto, Slovenia. Refer to Table 1 for function.

HPMC of the 2208 substitution type (HPMC 2208) and nominal viscosity of 15,000 mPa·s (cP), METOLOSE^®^ 90SH-15000SR QbD Samples, were obtained from Shin-Etsu Chemical Co., Ltd., Tokyo, Japan. Mixtures of up to three QbD samples from the obtained QbD sample kit were used in individual experiments to reach desired levels of HPMC FRCs according to the DoE. The producer’s certificate of analysis (CoA) results for individual QbD samples in the QbD sample kit, which are presented in Appendix A.

Lactose Monohydrate (Ph. Eur), FlowLac^®^ 100, was obtained from MEGGLE GmbH and Co. KG, Wasserburg, Germany. The same batch of FlowLac^®^ 100 was used in all the experiments. Refer to Table 1 for function.

Cellulose, Microcrystalline (Ph. Eur.), Avicel^®^ PH-102, was obtained from DuPont Nutrition Ireland, Little Island, Ireland. The same batch of Avicel^®^ PH-102 was used in all the experiments. Refer to Table 1 for function.

Silica, Colloidal Anhydrous (Ph. Eur.), i.e., Colloidal Silicon Dioxide (USP), AEROSIL^®^ 200 Pharma, was obtained from Evonik Operations GmbH, Essen, Germany. The same batch of AEROSIL^®^ 200 Pharma was used in all the experiments. Refer to Table 1 for function.

Magnesium stearate, Magnesium stearate EUR PHAR Vegetable, was obtained from FACI S.p.A., Carasco, Italy. The same batch of Magnesium stearate EUR PHAR Vegetable was used in all of the experiments. Refer to Table 1 for function.

Tablet composition is presented in Table 1. All the materials used complied with their Ph. Eur. specifications.

### 2.2. Methods

#### 2.2.1. DoE Generation

The chosen DoE, Central Composite Design (CCD), was generated in Minitab^®^ (v22.2.1). The DoE contains four factors, namely HPMC viscosity (HPMC_Visc), HPMC HP substitution (HPMC_HP), HPMC mean particle size (HPMC_PS), and the fraction of FlowLac^®^ 100 (Lac) in the FlowLac^®^ 100/Avicel^®^ PH-102 filler mixture. It contained a full factorial part of the DoE (16 experiments), part of the DoE with axial points, i.e., extreme levels of factors (8 experiments), and three central points (3 experiments). The order of experiments was randomized with the three central point experiments positioned in the beginning, the middle, and at the end of the DoE run. More details about the DoE are available in the Appendix A.

#### 2.2.2. Compression Mixture Preparation

The compression mixtures were prepared on a laboratory scale in 1 kg batch sizes using a 6 L biconical mixer at 32 rpm mixing speed. The volume fill % was between 1/3 and 2/3 of the available blender volume, dependent on the bulk volume of prepared compression mixtures. Homogenization of the ingredients before the addition of lubricant (magnesium stearate) was performed for 8 min, and additional homogenization after the addition of lubricant was performed for 2 min. Prior to each compression mixture preparation, selected HPMC QbD samples (up to 3 per experiment) were pre-homogenized in the same biconical mixer at 32 rpm for 8 min. The homogenization conditions were chosen based on preliminary experiments.

#### 2.2.3. Production of Tablets

Tablets were prepared via direct compression of compression mixtures using the Killian Pressima rotary tablet press and round, slightly concave (R = 30 mm) punches with a 12 mm diameter and bevelled edges. The same compression speed of 4800 tbl/h, the same fill-o-matic speed of 10 rpm, and the same main compression force of 20 kN (with no pre-compression) were used in all the performed experiments. The main compression force of 20 kN was selected based on preliminary experiments to achieve satisfactory friability results of tablets. The targeted average weight of tablets was 648 mg. Individual tablet weight variation was low, with a maximal observed relative standard deviation % (RSD%) of 0.86% and an average observed RSD% of just 0.54%.

#### 2.2.4. Carvedilol Release Profiles

The carvedilol release profiles were obtained using a method previously described by Košir et al. [22], which proved suitable for carvedilol (free base) release testing where carvedilol is incorporated into HPMC-based matrix tablets and was adequately discriminatory for studying the impact of HPMC FRC variability on carvedilol release. The method was not chosen with any intentional relation to a potential quality target product profile (QTPP), including the dosing interval. Therefore, a dissolution apparatus type 2 with paddles, paddle speed of 100 rpm, flow-through cuvettes, an autosampler, 900 mL of acetate buffer solution (pH = 4.5) per vessel, dissolution media temperature of 37 °C ± 0.5 °C, and sinkers to keep tablets at the bottom of the vessel were used. The amount of carvedilol released was spectrophotometrically determined at 285 nm from the measured absorbance and calculated using a calibration curve prepared in advance (carvedilol concentration range 0.00359–0.08985 mg/mL, achieved linear signal response with calibration curve RSQ of 0.99997). Four tablets per experiment were tested. The preselected time points for sampling were the same for all experiments and were as follows: start—0.5 h every 10 min, at 45 min—1 h–6 h every 30 min, and 6 h–24 h every 60 min. Carvedilol release at earlier time points (less than t = 10 min) was not analyzed because sampling the first data point at t = 10 min ensured a good signal-to-noise ratio in the utilized dissolution measurement system for all formulations. Sampling too early after tablet immersion could have led to carvedilol release data in which much of the observed variation arose from a low signal-to-noise ratio rather than genuine tablet-to-tablet differences.

Summary of experimental carvedilol release results is available in the Appendix A.

#### 2.2.5. Statistical Analysis of DoE Results and Generation of Models


*Generation of Models*


Statistical analysis of DoE results was conducted using Minitab^®^ (v22.2.1) in several stages. At each stage, multiple linear regression (MLR) models were generated separately for each time point (t = 10 min, t = 20 min, t = 30 min, t = 45 min, t = 1 h, t = 1.5 h, t = 2 h, …, t = 6 h, t = 7 h, …, t = 24 h). The initial (basic) predictor variables (*X* variables) in each stage were
**Lac** (fraction of FlowLac^®^ 100 in the FlowLac^®^ 100/Avicel^®^ PH 102 filler mixture)—**Factor A**,**HPMC_Visc** (HPMC viscosity)—**Factor B**,**HPMC_HP** (HPMC HP substitution)—**Factor C**, and**HPMC_PS** (HPMC mean particle size)—**Factor D**.

The distinction between individual stages of MLR model generation lay in the combination of model terms used for each time point. The full set of model terms for the CCD—i.e., the full response-surface methodology (RSM) model—included
(1)**Basic linear terms:**A (Lac)B (HPMC_Visc)C (HPMC_HP)D (HPMC_PS)(2)**Quadratic terms** (to model non-linear/curved relationships between *X* and *Y* variables using a second-order polynomial)**:**AA (Lac^2^)BB (HPMC_Visc^2^)CC (HPMC_HP^2^)DD (HPMC_PS^2^)(3)**Second-order interaction terms:**AB (Lac·HPMC_Visc)—interaction between Lac and HPMC_ViscAC (Lac·HPMC_HP)—interaction between Lac and HPMC_HPAD (Lac·HPMC_PS)—interaction between Lac and HPMC_PSBC (HPMC_Visc·HPMC_HP)—interaction between HPMC_Visc and HPMC_HPBD (HPMC_Visc·HPMC_PS)—interaction between HPMC_Visc and HPMC_PSCD (HPMC_HP·HPMC_PS)—interaction between HPMC_HP and HPMC_PS

The target *Y* variables for the MLR models were **the mean % carvedilol release** and **standard deviation (SD) of carvedilol release** at each time point. Separate MLR models were generated for each *Y* variable at every time point to uncover the dynamics of factors’ and interactions’ influence on carvedilol release throughout the carvedilol release profile.

In the first stage of modelling, **basic MLR models** were generated **using only the basic linear model terms (A, B, C, D)** for each time point. During this stage, mean % carvedilol release was selected as the target *Y* variable to facilitate a comparison of HPMC FRC’s influence on mean carvedilol release, with prior analyses using MLR and partial least squares regression (PLSR) performed by Košir et al. [22]. The SD of carvedilol release was incorporated as the target *Y* variable exclusively in MLR models generated during the second and third stages of model development.

In the second stage of modelling, **full RSM MLR models** were generated for each time point **using the complete set of CCD/RSM model terms (A, B, C, D, AA, BB, CC, DD, AB, AC, AD, BC, BD, CD)**. These models were developed for both target *Y* variables: mean % carvedilol release and SD of carvedilol release at each time point.

In the third stage of modelling, **optimized MLR models** were **generated using stepwise regression approaches** in Minitab^®^ **and selecting the best-performing models based on their predictive ability**. This stage involved two steps:(1)**Step 1: Model Optimization**For each time point, MLR models were optimized using all available stepwise regression methods in Minitab^®^’s DoE analysis module [59,60] Forward selection with Akaike Information Criterion (AIC),Forward selection with Bayesian Information Criterion (BIC),Stepwise regression (default α to enter and α to remove of 0.15 were used),Forward selection (default α to enter of 0.25 was used), andBackward elimination (default α of 0.10 was used).All **models were hierarchical** [60], meaning lower-order terms (e.g., linear terms A, B) were retained if their higher-order counterparts (e.g., quadratic AA or interaction AB) were included. For example:If the quadratic term AA (A^2^) was included, the linear term A was also retained.If the interaction term AB (A·B) was included, both linear terms A and B were retained.(2)**Step 2: Model Selection****Optimized MLR models were selected based on the highest predicted *R*^2^ (*R*^2^*_pred_*)** value (denoted as “R-sq(pred)” in Minitab^®^’s Model Summary output). These **models contained a hierarchical subset of terms from the full RSM MLR model for each time point,** maximizing predictive accuracy while avoiding overfitting.Interpretation of *R*^2^ and *R*^2^*_pred_* [61]: *R*^2^ (coefficient of determination): Proportion of variation in the target *Y* variable (mean % carvedilol release or SD of carvedilol release) explained by the model.*R*^2^*_pred_*: Measures a model’s ability to predict new observations by iteratively removing data points, re-estimating the model, and validating predictions. Higher *R*^2^*_pred_* values indicate better predictive performance.

A large discrepancy between *R*^2^ and *R*^2^*_pred_* suggests overfitting, where the model fits the sample data well but performs poorly on new data. This occurs when irrelevant terms are included, tailoring the model to noise rather than population effects [61].


*Interpretation of MLR Model Terms*


For the generated MLR models, *p*-values of model terms and their % contribution (derived from Minitab^®^’s Analysis of Variance (ANOVA) output) [62] were used to assess the strength of their effects on carvedilol release. In optimized MLR models generated using stepwise regression, the *p*-values should be interpreted as “descriptive statistics”, rather than as indicators of absolute statistical significance. They were not corrected using methods such as Bonferroni adjustments, because such corrections would be overly conservative and could hinder interpretation, especially given the known effects of the studied factors on drug release from HPMC matrix tablets, including carvedilol, reported in the literature. Instead of strictly adhering to formal statistical corrections, a more pragmatic approach was employed: The statistical indicators of effect size and significance were interpreted in the context of prior knowledge about the impact of the studied factors on drug release. This pragmatic approach, which incorporates prior mechanistic knowledge, was also applied when evaluating the variable selection results from the stepwise regression.


*Statistical Significance (p-Values)*


Traditionally, a model term is considered statistically significant at a significance level (*α*) of 0.05 if its *p*-value is ≤0.05 [62]. However, in this study, *p*-values were analyzed dynamically across time points to evaluate how the influence of terms on drug release evolved over time, rather than solely categorizing terms as “significant” or “non-significant.”


*Contribution % (Sequential Sums of Squares)*


The contribution % of each model term represents the proportion of the total variation (sequential sums of squares, Seq SS) explained by that term. Seq SS quantifies the unique variation attributed to a term after accounting for previously included terms [62]. Key points:The sum of all contribution % equals the model’s *R*^2^ (coefficient of determination) expressed as a percentage, reflecting the proportion of variation in the target *Y* variable (mean % carvedilol release or SD of carvedilol release) explained by the model.The remaining unexplained variation is attributed to the model’s error.

Similarly to *p*-values, contribution % was analyzed dynamically across time points to reveal how the relative importance of terms evolved during drug release.


*Main Effects and Enteraction Effects*


The influence of main effects (individual factors) and interaction effects (combined factors) on carvedilol release was analyzed using Minitab^®^’s main effects plots and interaction plots for each time point. In main effects plots, the slope of the lines indicates the direction (positive or negative) and intensity (steepness of the slope) of a factor’s effect on carvedilol release. A steeper slope (in either direction) corresponds to a stronger influence of the factor. Interaction plots visualize second-order interactions between two factors. One factor is displayed at three levels (e.g., low, medium, high), while the other factor’s relationship with the target *Y* variable (mean % carvedilol release or SD of carvedilol release) is shown across these levels. Non-parallel lines indicate an interaction effect. The steepness and direction of the slopes reflect the strength and nature of the interaction—steeper divergences indicate a stronger interaction effect, while parallel lines indicate no interaction [63,64,65,66].

More detailed information on the generated MLR models for mean % and SD of carvedilol release at each time point is provided in the Appendix A, and Minitab^®^ reports.

#### 2.2.6. In-Process Control Testing of Tablets

In-process control (IPC) testing of tablets was performed for informational purposes only. Tablet mass, hardness, and thickness were analyzed using an ERWEKA MultiCheck (ERWEKA GmbH, Langen, Germany). The target mass of all tablets was the same across all experiments. As the main compression force was set to 20 kN (with no pre-compression) and the compression speed and fill-o-matic speed were identical in all experiments, differences in hardness and thickness mainly arose from formulation variations according to the DoE. The effect of tablet mass variation was negligible, as mass variation within and between experiments was very low. The IPC results for representative samples from individual experiments are summarized in Appendix A. Tablet mass was measured on 20 tablets per experiment, while hardness and thickness were measured on 10 tablets per experiment.

Tablet friability was tested using a USP/PhEur/JP-compliant ERWEKA TAR friability and/or abrasion tester (ERWEKA GmbH, Langen, Germany). For these tests, the friability drum was operated at 25 rpm for 4 min, using a representative sample of approximately 6.5 g of tablets per experiment. Results are presented in Appendix A. Friability was generally low, below 0.1% in all experiments.

## 3. Results and Discussion

### 3.1. Overview

Experiments using varying levels of the selected main HPMC FRCs (HPMC_Visc, HPMC_HP, HPMC_PS) and different fractions of lactose FlowLac^®^ 100 within the lactose FlowLac^®^ 100/MCC Avicel^®^ PH 102 filler mixture (Lac) yielded significant variation in both mean carvedilol release and the SD of carvedilol release, as shown in Figure 1 (see also Appendix A for a more detailed data representation). It is evident that a higher content of water-soluble lactose in tablets results in significantly faster carvedilol release profiles and in increased carvedilol release variation. This was expected from our previous study, which showed that using MCC as a filler instead of lactose results not just in slower carvedilol release profiles but also in lower intrabatch carvedilol release variation [53]. Further, intrabatch carvedilol release variation also arises from mixing two different fillers in the tablets, as well as the inclusion of up to three HPMC QbD samples in each experiment.

The strategy of blending up to three HPMC QbD samples per experiment (HPMC FRC data for these samples are provided in Appendix A) enabled the DoE factor levels to closely approximate the theoretical targets (see Appendix A). However, this approach involved trade-offs. It narrowed the achievable range between the maximum and minimum HPMC FRC levels across experiments and introduced additional intrabatch variability in carvedilol release due to the mixing of HPMC QbD samples within individual experiments. Blending HPMC QbD samples in individual experiments was necessary to achieve the desired DoE factor levels, as HPMC QbD samples with the required FRC levels could not otherwise be obtained. The authors acknowledge that the carvedilol release variability resulting from blending HPMC QbD samples is therefore confounded with variability arising from differences in average HPMC FRC levels. This approach also assumes linear blending behaviour of HPMC properties. Furthermore, carvedilol release testing was performed on only four tablets per experiment due to analytical resource limitations, which restricts the accuracy of estimating both the mean and SD of carvedilol release—particularly the SD. Therefore, especially regarding the influence of HPMC FRCs on SD, these results should be treated with caution and considered preliminary. In future research, a better approach would be to obtain or manufacture HPMC QbD samples with FRC levels tailored to the factor levels in the DoE. Additionally, analyzing more tablets per experiment (e.g., 12 or 24) would improve the accuracy of mean and SD estimates of drug release.

In the following sections, the influence of selected HPMC FRCs and Lac on carvedilol release is analyzed in terms of their main effects and interactions. While the analysis of mean carvedilol release is likely more statistically robust, the evaluation of carvedilol release variability (SD of carvedilol release) in release profiles warrants caution. Firstly, only four tablets were analyzed per experiment. Secondly, the blending of HPMC QbD samples to achieve target HPMC FRC levels introduces confounding variation. This confounding effect arises because variability from the experimental design (e.g., blending of HPMC QbD samples within individual experiments) overlaps with variability attributed to the studied factors (HPMC FRCs and Lac). Nevertheless, despite these limitations, the data were analyzed comprehensively to extract maximum insights. This pragmatic approach ensures that all available information—including trends in intrabatch carvedilol release variability—is leveraged to assess the influence of the selected factors.

The influence of the selected factors on mean carvedilol release was interpreted using the generated MLR models for time points from t = 20 min to t = 16 h; by 16 h, over 75% of carvedilol had been released in all runs. The *R*^2^*_pred_* at t = 10 min was significantly lower than at t = 20 min and successive time points (see Appendix A). Furthermore, after 16 h, one experiment reached a plateau (see Figure 1 and Appendix A), reducing the predictive accuracy of MLR models for later time points and hindering the analysis of factors’ and interactions’ impact on carvedilol release. For the basic MLR models predicting mean carvedilol release, the most relevant information on model terms is presented in Appendix A, and the main effects plots in Appendix A. Considering the range from t = 20 min to t = 16 h, Appendix A are particularly relevant. Using these basic models, only the main effects of the selected factors on mean carvedilol release were studied. Optimized MLR models for predicting mean carvedilol release are detailed in Appendix A (model terms) and Appendix A (main effects and interaction plots). For the t = 20 min to t = 16 h range, Appendix A are relevant. These optimized models allow for studying not only the main effects but also interactions between Lac and certain HPMC FRCs, and interactions between certain HPMC FRCs.

For interpreting the selected factors’ influence on the SD of carvedilol release, the range of MLR models from t = 20 min to t = 14 h was used. The same starting point (t = 20 min) as for the mean release analysis was used for consistency. However, the predictive ability of MLR models for SD of carvedilol release after t = 14 h was significantly reduced (see Appendix A). Overall, optimized MLR models for SD showed substantially lower *R*^2^*_pred_* than those for mean release (Appendix A vs. S15–S16), indicating greater randomness in variability. For the optimized MLR models predicting the SD of carvedilol release, relevant model terms are presented in Appendix A, and main effects/interaction plots in Appendix A. For the t = 20 min to t = 14 h range, Appendix A are relevant. These optimized models allow for investigating the main effects of selected factors on SD of carvedilol release, as well as the influence of potential interactions between Lac and certain HPMC FRCs, and interactions among HPMC FRCs on SD of carvedilol release.

### 3.2. Interpretation of Basic MLR Models Predicting Mean Carvedilol Release in Individual Time Points

The dynamics of factors’ influence on mean carvedilol release, evident from Figure 2 and Figure 3, and Appendix A, demonstrates that Lac has a much larger impact on carvedilol release than the selected HPMC FRCs. The main effect of Lac keeps increasing throughout the relevant range for interpretation of selected factors’ influence on mean carvedilol release (t = 20 min to t = 16 h). The nature of Lac’s impact on carvedilol release, evident from main effects plots in Appendix A, is consistent with previous literature, that is, that the usage of water-soluble fillers results in faster drug release than the usage of water-insoluble ones—a higher fraction of water-soluble FlowLac^®^ 100 in the filler mixture results in a faster mean carvedilol release.

Among the HPMC FRCs, HPMC_PS dominates early release but declines to a minimum by ≈ 10 h (Figure 2, Figure 3, and Appendix A). As described in the Section 1 of the article and according to literature, larger HPMC particles result in faster carvedilol release. HPMC_Visc and HPMC_HP have minimal early effects, then gradually rise to plateaus—HPMC_Visc by ≈ 3 h and HPMC_HP by ≈ 7 h (Figure 2, Figure 3, and Appendix A). As mentioned in the Section 1 and in line with referenced literature, higher viscosity of HPMC decreases the rate of carvedilol release, while higher HP substitution increases it (Appendix A). Importantly, each HPMC factor’s impact shifts dynamically through the release profile (Figure 2, Figure 3, Appendix A, and Appendix A). These observations are generally in line with previously conducted research by Košir et al. [22] where HPMC 2208 K4M was used instead of HPMC 2208 K15M and lactose was solely used as a filler instead of the lactose/MCC mixture.

### 3.3. Interpretation of Optimized MLR Models Predicting Mean % of Carvedilol Release in Individual Time Points

Analysis of the optimized MLR models predicting mean carvedilol release revealed further insights into the roles of HPMC FRCs and the lactose/MCC ratio. As with the basic models, Lac exerts the strongest effect from the outset, and its influence grows steadily over the analysis window (t = 20 min to 16 h) (Figure 4 and Figure 5).

Early in the release (up to ≈5 h), the interaction terms HPMC_Visc·HPMC_PS and Lac·HPMC_Visc contribute more to mean release than the main effect of HPMC_Visc itself (Figure 4, Figure 5, and Appendix A), illustrating the main effects can have little meaning when they are involved in significant interactions [66]. Although included in all models to preserve hierarchy, HPMC_Visc’s standalone impact is relatively minor until after ≈5 h.

HPMC_HP appears only as a main effect—absent from significant interactions—and becomes meaningful only after ≈7 h, persisting through the 16 h endpoint (Figure 4, Figure 5, and Appendix A).

HPMC_PS shows a pronounced main effect at early time points, diminishing to a minimum by ≈10 h, paralleling the basic MLR models’ observations (Figure 2, Figure 3, and Appendix A). Notably, the HPMC_Visc·HPMC_PS interaction—which includes HPMC_PS—accounts for a larger contribution % than HPMC_PS alone and, while it too tapers toward 16 h, it remains more influential than HPMC_PS across the entire period (t = 20 min to 16 h) (Figure 5 and Appendix A).

Optimized MLR models’ analysis predicting mean carvedilol release—from the viewpoint of dynamic changes in HPMC FRCs’ and Lac’s impact on mean carvedilol release, including the mentioned interactions—reveals three distinct phases:Phase 1 (Appendix A):○characterized by the non-typical or reversed impact of HPMC viscosity and the absence of HPMC HP substitution’s impact on mean carvedilol release○duration from t = 20 min (start of relevant optimized MLR models’ interpretation range) to ≈2 h○probably the phase of initial HPMC hydration and swelling○up to app. between 20% and 40% of carvedilol is released during this phase (Figure 1, Appendix A)Phase 2 (Appendix A):○characterized by the typical (in line with the literature mentioned in the Section 1 of the article) impact of HPMC viscosity and the absence of HPMC HP substitution’s impact on mean carvedilol release○duration from ≈2.5 h to ≈6 h○phase of established control over carvedilol release where the influence of particulate HPMC properties (such as HPMC_PS) and HPMC molecular properties, particularly HPMC_Visc, intertwine in a complex way○up to app. between 40% and 65% of carvedilol is released during this phase (Figure 1, Appendix A)Phase 3 (Appendix A):○characterized by the presence of HPMC HP substitution’s impact on mean carvedilol release○duration from ≈7 h to ≈16 h (end of relevant optimized MLR models’ interpretation range)○phase where at the beginning of this phase, the influence of HPMC_PS in terms of its main effect falls to a minimum and where the molecular HPMC properties, such as HPMC_Visc and HPMC_HP, show a dominant impact on mean carvedilol release○up to app. upwards of 75% of carvedilol is released during this phase (Figure 1, Appendix A)


In all of the mentioned phases, Lac has the strongest impact on mean carvedilol release, HPMC_PS’s impact in terms of its main effect is present (although it has different contributions to the mean carvedilol release, particularly a lesser impact in most of Phase 3 of mean carvedilol release), and both mentioned interactions—HPMC_Visc·HPMC_PS and Lac·HPMC_Visc—play an important role throughout all of the phases of mean carvedilol release, although their contribution changes somewhat among these phases.


*Phase 1 of Mean Carvedilol Release*


Phase 1 of mean carvedilol release is primarily characterized by an atypical effect of HPMC_Visc, opposite to the trend reported in the literature (Figure 6). This phase extends from t ≈ 20 min (the beginning of the relevant interpretation range for the optimized MLR models) to t ≈ 2 h (Appendix A). It likely corresponds to the initial hydration and swelling of HPMC, during which the polymer network establishes control over carvedilol release and influences the burst-release behaviour. Higher lactose and lower MCC levels in the lactose–MCC mixture (Lac) accelerate initial carvedilol release, consistent with prior studies on filler effects [53].

HPMC_PS’s main effect on mean carvedilol release (Figure 6 and Appendix A) aligns with literature cited in the introduction. HPMC_HP has no influence, neither as a main effect nor in interactions with other HPMC FRCs (viscosity, particle size) or Lac. Notably, HPMC_Visc’s main effect warrants discussion: analysis of optimized MLR models indicates that higher HPMC_Visc increases mean carvedilol release in Phase 1 (Figure 6 and Appendix A), contrary to literature expectations. This may result from higher-viscosity HPMC hydrating and swelling more slowly [11,29,30], delaying sufficient control over carvedilol release, despite its higher water uptake capacity [11,31] and stronger, more erosion-resistant gel with increased diffusion-path tortuosity [1,2,3,10,20,21,23,26,27]. Lower surface hydrophilicity of higher-viscosity HPMC may also slow initial wetting, hydration, and swelling [28].

Significant interactions—the interaction between Lac and HPMC_Visc (Lac·HPMC_Visc), and the interaction between HPMC_Visc and HPMC_PS (HPMC_Visc·HPMC_PS) (Figure 7 and Appendix A)—are present across all three phases of mean carvedilol release, from t = 20 min to t = 16 h (Appendix A), with consistent effects. Detailed here for Phase 1, they are referenced in Phases 2 and 3.

The Lac·HPMC_Visc interaction indicates that at lower HPMC viscosity, increasing lactose (decreasing MCC) in the filler mixture markedly increases mean carvedilol release rate compared to higher viscosity (left side of Figure 7). Lower viscosity reduces diffusion-path tortuosity and increases gel erosion due to a weaker gel layer [1,2,3,10,20,21,23,26,27]. Water-soluble excipients like lactose enhance water transport via osmotic pressure, dilute the gel layer, reduce tortuosity, and accelerate drug diffusion and erosion [1,2,42,43,48,50,52]. Conversely, water-insoluble MCC strengthens the gel layer, resisting erosion and increasing tortuosity [42,45,52]. The Lac·HPMC_Visc interaction could also be inspected via different levels of Lac (top of Figure 7). At low Lac levels, increasing HPMC viscosity increases release rate, while at high Lac levels, it decreases release rate (top of Figure 7). Lactose’s osmotic effect may “salt in” HPMC, aiding hydration and swelling, akin to the action of some salt-form drugs or added salts in HPMC matrix tablets [2]. At low Lac, higher-viscosity HPMC hydrates slowly [11,29,30], and insufficient lactose content limits “salting in” of HPMC, increasing release. At high Lac, sufficient lactose content enhances HPMC hydration via its “salting-in” effect, allowing higher-viscosity HPMC to slow release by increasing gel tortuosity and reducing its erosion. Thus, higher HPMC viscosity requires higher lactose content for efficient hydration and swelling to control carvedilol release.

The HPMC_Visc·HPMC_PS interaction shows that at smaller HPMC particle size, increasing viscosity decreases release rate, which is consistent with literature (bottom of Figure 7). At larger particle sizes, increasing viscosity increased release rate, likely due to slower swelling of larger particles [1,2,26] and higher-viscosity HPMC [11,29,30], reducing control over release. At low HPMC viscosity, increasing its particle size decreases release rate (right side of Figure 7), possibly because small particles with low viscosity cause rapid polymer relaxation and erosion, and larger particles slightly slow hydration, enhancing control over carvedilol release (Figure 7 and Appendix A).


*Phase 2 of Mean Carvedilol Release*


Phase 2 of mean carvedilol release, spanning from t ≈ 2.5 h to t ≈ 6 h, differs from Phase 1 primarily by HPMC_Visc exhibiting a typical main effect, consistent with literature cited in the Introduction: higher HPMC viscosity reduces mean carvedilol release rate (Figure 8 and Appendix A). This typical effect suggests that carvedilol release is controlled by both molecular HPMC characteristics (e.g., viscosity, related to polymer chain length) and particulate characteristics (e.g., mean particle size). However, molecular characteristics are less dominant than in Phase 3, as HPMC_HP does not significantly influence carvedilol release in this phase as it does in Phase 3. HPMC_PS’s main effect begins to subside, while HPMC_Visc’s main effect increases (Figure 5, Appendix A), indicating that both particulate and molecular HPMC characteristics are significant in this phase. As in Phase 1, Lac exerts the strongest impact on mean carvedilol release, with its effect continuing to increase and surpassing Phase 1 levels (Figure 5, Appendix A). The interactions Lac·HPMC_Visc and HPMC_Visc·HPMC_PS, detailed in Phase 1, remain significant (Figure 9 and Appendix A). The Lac·HPMC_Visc interaction achieves statistical significance (*p* < 0.05; Figure 4, Appendix A) by conventional statistical standards [62], though its contribution % declines slightly from initial levels (Figure 5, Appendix A). The HPMC_Visc·HPMC_PS interaction’s contribution % decreases gradually but remains statistically significant (Figure 4, Appendix A).


*Phase 3 of Mean Carvedilol Release*


Phase 3 of mean carvedilol release, spanning from t ≈ 7 h to t ≈ 16 h, contrasts with Phases 1 and 2 by introducing HPMC_HP’s main effect (Figure 10 and Appendix A). Higher HPMC_HP increases mean carvedilol release rate, consistent with literature cited in the introduction (Figure 10 and Appendix A). HPMC_HP influences mean carvedilol release solely as a main effect, without interactions with other HPMC FRCs (viscosity, particle size) or Lac, and is statistically significant by conventional statistical standards (Figure 4, Appendix A). This does not imply that HPMC_HP has no effect on carvedilol release prior to t ≈ 7 h; rather, its impact is not identified as significant using the methodology of generating optimized MLR models via stepwise regression in Minitab^®^ and selecting the best-performing models based on their predictive ability. The same conclusion applies to the impact of all HPMC FRCs throughout the carvedilol release profile. Lac continues to exert the strongest impact, with its contribution % increasing (Figure 5, Appendix A). HPMC_Visc’s main effect peaks in this phase (Figure 5, Appendix A) but does not achieve statistical significance (Figure 4, Appendix A). HPMC_PS’s main effect diminishes significantly at the phase’s onset, retaining residual impact throughout (Figure 4, Figure 5, Appendix A). The prominence of HPMC_Visc and HPMC_HP, alongside HPMC_PS’s reduced effect, indicates that molecular characteristics of HPMC dominate over particulate characteristics in controlling mean carvedilol release. The HPMC_Visc·HPMC_PS interaction (Figure 11 and Appendix A) becomes statistically non-significant by conventional standards (Figure 4, Appendix A) but retains non-negligible contribution (Figure 5, Appendix A). The Lac·HPMC_Visc interaction shown in Figure 11 and Appendix A remains statistically significant (Figure 4, Appendix A) and contributes substantially (Figure 5, Appendix A).

These dynamics across all three phases likely relate to front movements—swelling, diffusion, and erosion fronts [1,2,3,10,12,13,17]. As most of the tablet is hydrated and swollen, the swelling front’s inward movement is limited, enhancing the influence of HPMC’s molecular characteristics (e.g., viscosity, HP substitution) on mean carvedilol release, while HPMC_PS’s main effect nearly diminishes. The persistent HPMC_Visc·HPMC_PS interaction indicates ongoing particle swelling, and the Lac·HPMC_Visc interaction suggests residual lactose aids hydration of remaining dry HPMC.

### 3.4. Interpretation of Optimized MLR Models Predicting SD of Carvedilol Release in Individual Time Points

Analysis of optimized MLR models predicting the standard deviation (SD) of carvedilol release provides insight into the influence of HPMC FRCs (viscosity, particle size, HP substitution) and Lac on SD of carvedilol release. The variability in carvedilol release due to HPMC FRCs is partially confounded by the use of mixed HPMC QbD samples in individual experiments. The extent of this impact on result interpretation is unclear, warranting caution. Additionally, the predictive ability of SD models is significantly lower than that of mean carvedilol release models, explaining only 25–40% of SD variation and containing more random error (Appendix A vs. Appendix A). The research results indicating the influence of HPMC FRCs on SD of carvedilol release should generally be treated with caution and viewed as preliminary due to having several drawbacks in the utilized methodology, as already mentioned in Section 3.1. Despite analyzing four tablets per experiment, limited information on intrabatch carvedilol release variability was obtained. To fully utilize the DoE and carvedilol release results, MLR models predicting SD of carvedilol release were generated, enabling model-dependent analysis of HPMC FRCs’ and Lac’s influence on SD. As with mean release models, Lac exerts the most significant impact on SD of carvedilol release within the relevant interpretation range (t = 20 min to t = 14 h), but its effect is non-linear, as indicated by a quadratic term in the MLR models (Figure 12, Figure 13 and Appendix A). Initially, the Lac·HPMC_Visc interaction and HPMC_Visc’s main effect were significant, but their impact diminished after t = 1.5 h (Figure 12, Figure 13 and Appendix A); HPMC_Visc’s main effect reappears from t = 10 h to t = 14 h (Figure 12, Figure 13 and Appendix A). Non-linear effects of HPMC_PS and HPMC_HP appear at t = 30 min and t = 45 min (Figure 12, Figure 13, Appendix A), possibly an anomaly due to mixing HPMC QbD samples to achieve target DoE levels. From t = 3.5 h onward, HPMC_HP is significant as a main effect and in the Lac·HPMC_HP interaction, which was not observed in mean carvedilol release models (Figure 12, Figure 13 and Appendix A).

Optimized MLR models’ analysis predicting SD of carvedilol release from the viewpoint of the dynamic changes in HPMC FRCs’ and Lac’s impact on SD of carvedilol release, including the mentioned interactions, reveals four distinct phases:
Phase 1 (Appendix A): ○characterized by the non-linear impact of Lac as a main effect, linear impact of HPMC_Visc as a main effect, and the impact of their interaction (Lac·HPMC_Visc) on SD of carvedilol release○duration from t = 20 min (start of relevant optimized MLR models’ interpretation range) to ≈1.5 h○probably the phase of initial HPMC hydration and swelling○up to app. between 20% and 35% of carvedilol is released during this phase (Figure 1, Appendix A)Phase 2 (Appendix A):○characterized solely by the non-linear impact of Lac as a main effect on SD of carvedilol release○duration from ≈2 h to ≈3 h○probably the phase of initial HPMC hydration and swelling○up to app. between 28% and 50% of carvedilol is released during this phase (Figure 1, Appendix A)○this phase can be viewed as an extension of Phase 1 or a prelude to Phase 3 as it is a short intermediate phase
Phase 3 (Appendix A):○characterized by the non-linear impact of Lac as a main effect, linear impact of HPMC_HP as a main effect and the impact of their interaction (Lac·HPMC_HP) on SD of carvedilol release○duration from ≈3.5 h to ≈9 h○phase in which hydration and swelling of HPMC plays an important role; however, the molecular characteristics of HPMC, particularly the HPMC_HP, are becoming increasingly important as factors influencing intrabatch carvedilol release variation○up to app. between 50% and 85% of carvedilol is released during this phase (Figure 1, Appendix A)Phase 4 (Appendix A):○characterized by the non-linear impact of Lac as a main effect, linear impact of HPMC_Visc as a main effect, linear impact of HPMC_HP as a main effect, and the interaction between Lac and HPMC_HP (Lac·HPMC_HP) on SD of carvedilol release○duration from ≈10 h to ≈14 h (end of relevant optimized MLR models’ interpretation range)○phase in which molecular characteristics of HPMC such as HPMC_Visc and HPMC_HP, together with Lac, show a dominant impact on SD of carvedilol release○up to app. between 68% and 95% of carvedilol is released during this phase (Figure 1, Appendix A)


The identified phases of SD of carvedilol release do not fully align with the recognized phases of mean carvedilol release, although some similarities are present. In both cases, the phases were determined based on changes in the model terms of optimized MLR models, which yielded the highest predictive performance at each time point. The estimation of the SD of carvedilol release, based on four tablets per experiment, is likely less accurate than the estimation of the mean release, which may partially explain the observed discrepancy. Furthermore, there is no conclusive evidence in the published literature, suggesting that the mean release rate and the SD of carvedilol (or any other drug) should be perfectly correlated. This may account for the observation of distinct phases and different subsets of optimal model terms, providing maximal predictive ability for the mean and SD of carvedilol release, respectively. Additionally, during phases 3 and 4 of SD of carvedilol release, the interaction term Lac·HPMC_HP was identified as important for predicting the SD, whereas it was not recognized as significant in any phase of the mean carvedilol release.


*Phase 1 of SD of Carvedilol Release*


Phase 1 of SD of carvedilol release is characterized by the non-linear main effect of Lac, the linear main effect of HPMC_Visc, and their interaction on SD of carvedilol release, as shown in Figure 14 and Figure 15, spanning from t = 20 min (start of relevant optimized MLR models’ interpretation range) to t ≈ 1.5 h (Appendix A). Lac’s non-linear effect suggests an optimal lactose-to-MCC ratio in the filler mixture of approximately 43:57, near 1:1, minimizing carvedilol release variability (Figure 14). This may reflect optimal mixing for tablet-to-tablet reproducibility or an optimal lactose level that enhances HPMC hydration via a “salting-in” effect, enabling efficient swelling to control carvedilol release. Suboptimal lactose levels may reduce “salting in” of HPMC, impairing HPMC hydration, while excessive lactose may “salt out” HPMC, competing for water and suppressing hydration [36]. This non-linear Lac effect persists across all four phases and is referenced in subsequent phases. The “salting out” of HPMC has been described as an effect caused by salts—either present in the dissolution media or within the tablets—that have a high affinity for water. These salts can dehydrate HPMC by “stealing” water from the gel layer, thereby weakening its structure or even preventing the effective formation of the gel layer in the first place [2].

HPMC_Visc’s linear effect, where higher viscosity reduces variability, is straightforward, as increased viscosity enhances diffusion-path tortuosity and forms a stronger, erosion-resistant gel [1,2,3,10,20,21,23,26,27]. This effect reappears in Phase 4.

The Lac·HPMC_Visc interaction, shown in Figure 15 (also Appendix A), is notable. The lower left part of Figure 15 indicates that optimal Lac levels vary with HPMC_Visc: higher-viscosity HPMC requires more lactose to minimize release variability. Lower-viscosity HPMC hydrates and swells faster [11,29,30], potentially due to higher surface hydrophilicity [28], needing less lactose for “salting in” of HPMC, while higher-viscosity HPMC hydrates more slowly [11,29,30], requiring more lactose. Optimal lactose levels are non-linear: insufficient lactose limits “salting in” of HPMC, while excess lactose induces a “salting-out” effect. The upper right part of Figure 15 shows that at low Lac levels, increasing HPMC_Visc increases variability due to slower hydration (and consequently swelling) [11,29,30], reducing control over release of carvedilol. At medium to high Lac levels, “salting in” of HPMC enhances HPMC hydration (and consequently swelling), and higher HPMC_Visc reduces variability by increasing tortuosity and gel strength. This Lac·HPMC_Visc interaction persists in Phases 3 and 4 and is referenced in subsequent phases.


*Phase 2 of SD of Carvedilol Release*


Phase 2 of SD of carvedilol release is characterized solely by the non-linear impact of Lac (Figure 16 and Appendix A), as detailed in Phase 1. It may be considered an extension of Phase 1 or a prelude to Phase 3, as it is a brief intermediate phase spanning from t ≈ 2 h to t ≈ 3 h.


*Phase 3 of SD of Carvedilol Release*


Phase 3 of SD of carvedilol release is characterized by the non-linear main effect of Lac, the linear main effect of HPMC_HP, and their interaction on SD of carvedilol release, as shown in Figure 17 and Figure 18 (Appendix A), spanning from t ≈ 3.5 h to t ≈ 9 h. Lac’s non-linear effect, consistent across all four phases, was detailed in Phase 1. HPMC_HP’s linear effect indicates that higher HP substitution increases carvedilol release variability. Increased HP substitution enhances polymer relaxation and erosion, contributing to the erosion release mechanism and reducing diffusion-path tortuosity, leading to faster release [36] and reduced control, thus increasing variability (Figure 17 and Appendix A). HP substituents act as spacers between polymer chains during H-bond formation, promoting relaxation and facilitating diffusion.

The Lac·HPMC_HP interaction (Figure 18 and Appendix A) is partially analogous to the Lac·HPMC_Visc interaction in Phase 1 (Figure 15 and Appendix A). Higher HPMC_HP increases hydrophilicity of HPMC, accelerating hydration and swelling, which are critical for controlling drug release [35]. The lower left part of Figure 18 shows that optimal Lac levels vary with HPMC_HP: lower HP substitution, with reduced hydrophilicity, requires higher lactose for a pronounced “salting-in” effect to enhance hydration. Conversely, higher HP substitution needs less lactose for optimal control over variability. Insufficient lactose content results in an inadequate “salting-in” effect, reducing control and increasing variability, while excessive lactose induces “salting out” of HPMC, also increasing variability. From the perspective of Lac levels (top right part of Figure 18), low Lac levels limit “salting in” of HPMC, but increasing HPMC_HP enhances hydrophilicity (and consequently hydration and swelling), improving control and reducing variability. At higher Lac levels, increased HPMC_HP excessively promotes relaxation, reducing control and increasing variability. This interaction persists in Phase 4, where it is also referenced.


*Phase 4 of SD of Carvedilol Release*


This phase introduces no new main effects or interactions but is characterized by both molecular characteristics of HPMC (HPMC_Visc and HPMC_HP) as main effects, alongside the consistent non-linear main effect of Lac (Figure 19 and Appendix A). The Lac·HPMC_HP interaction, present in Phase 3, persists (Figure 20 and Appendix A). Spanning from t ≈ 10 h to t ≈ 14 h, this phase indicates that HPMC’s molecular characteristics dominate in controlling carvedilol release variability alongside Lac.

**Figure 18 pharmaceutics-17-01358-f018:**
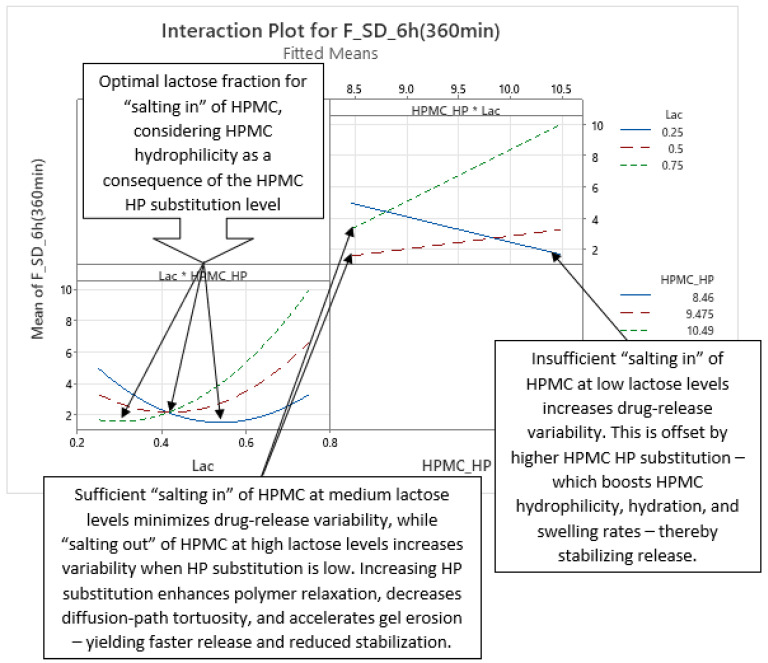
Representative example of an interaction influencing the SD of carvedilol release—the interaction between HPMC HP substitution (HPMC_HP) and the lactose fraction in the filler mixture (Lac). This interaction is shown here for t = 6 h as an example for Phase 3 of SD of release (more examples from Phase 3 are available in Appendix A). The same interaction is also present in Phase 4 of SD of release (see Figure 20). An asterisk between the names of each pair of factors shown in the figure represents the factors’ second-order in-teraction.

**Figure 19 pharmaceutics-17-01358-f019:**
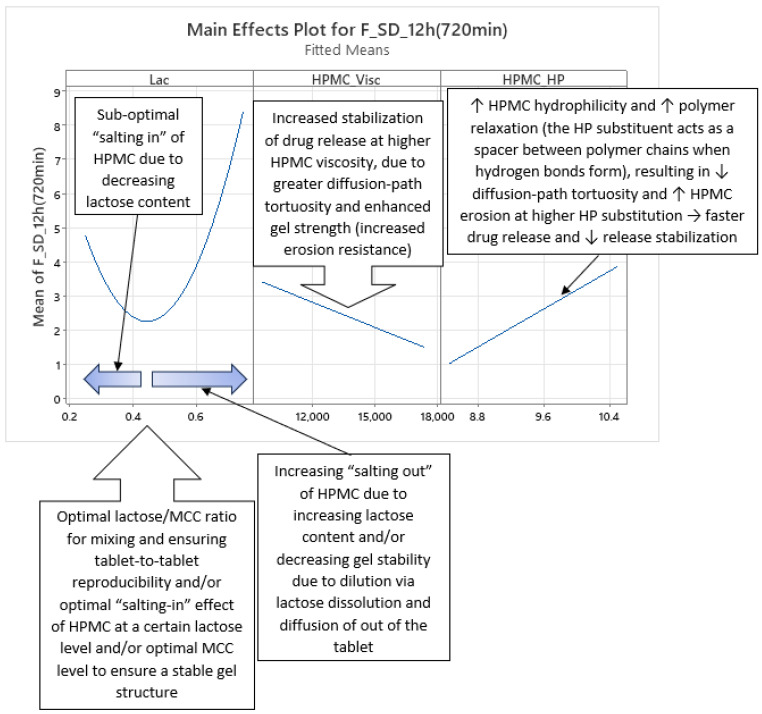
Representative example of main effects influencing SD of carvedilol release in Phase 4 of SD of release. Data are shown for t = 12 h, though the direction of these effects remained consistent across the timeframe t = 10 h to 14 h (Appendix A).

**Figure 20 pharmaceutics-17-01358-f020:**
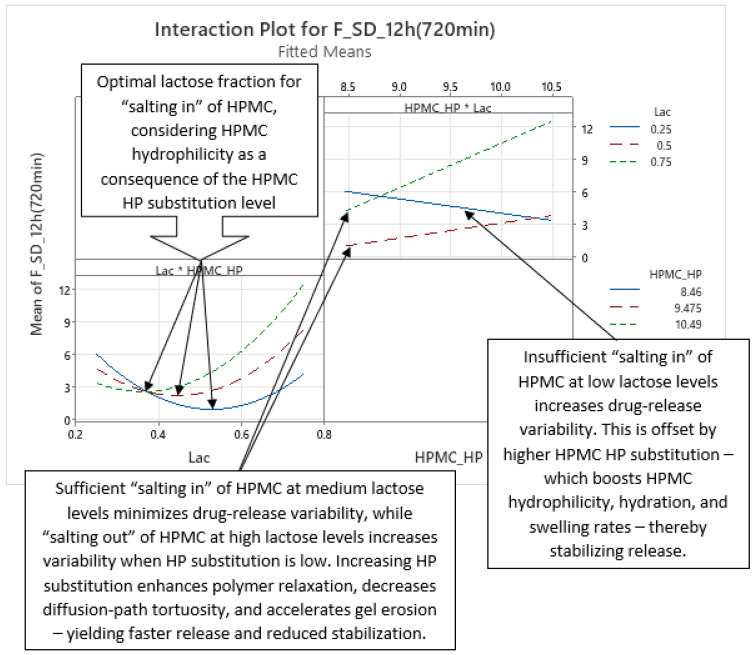
Representative example of an interaction influencing the SD of carvedilol release—the interaction between HPMC HP substitution (HPMC_HP) and the lactose fraction in the filler mixture (Lac). This interaction is shown here for t = 12 h as an example for Phase 4 of SD of release (more examples from Phase 4 are available in Appendix A). The same interaction is also present in Phase 3 of SD of release (see Figure 18). An asterisk between the names of each pair of factors shown in the figure represents the factors’ second-order in-teraction.

### 3.5. Recap of Complex Relationships Within HPMC FRCs and Lactose/MCC Ratio in the Filler Mixture Influencing Mean Carvedilol Release and Release Variability

The use of a high-resolution CCD DoE, incorporating selected HPMC FRCs (viscosity, HP substitution, particle size) and Lac as factors, provided critical insight into how HPMC FRCs and Lac modulate drug release. By frequently analyzing multiple tablets per experiment and developing optimized, interpretable MLR models for each time point, rather than using time as an *X* variable, this approach enabled dynamic analysis of factor effects on mean carvedilol release and its variability. This revealed complex interactions influencing both mean release and variability. In summary, Figure 21 illustrates how studied factors and their second-order interactions, uncovered through CCD and HPMC QbD sample mixtures to achieve approximately targeted DoE levels, influences mean carvedilol release. Similarly, Figure 22 depicts the factors and interactions affecting carvedilol release variability.

## 4. Conclusions

This study provided new insight into the complex interactions of selected HPMC FRCs (viscosity, HP substitution, particle size) and lactose/MCC ratio in the filler mixture in modulating carvedilol release from directly compressed HPMC K15M matrix tablets. Mixing HPMC QbD samples to achieve target DoE levels in a CCD, despite some limitations, enabled identification of key second-order interactions and confirmed established main effects influencing drug release. Optimized MLR models with hierarchical term subsets, offering high predictive ability compared to excessively overfitted full RSM models, were developed for each tested time point, revealing the dynamics of factors and interactions affecting carvedilol release. This approach was applied to both mean carvedilol release and its variability, an often-overlooked aspect of HPMC matrix tablet studies. Results suggest that tailoring lactose/MCC ratio in the filler mixture to HPMC FRC levels can partially mitigate the impact of HPMC FRC batch-to-batch variation on interbatch release variability. However, complete compensation is not feasible within the described formulation space, as HPMC FRCs exert distinct dynamic influences and interactions within themselves and with the filler mixture (and potentially other excipients) throughout the release profile.

## Figures and Tables

**Figure 1 pharmaceutics-17-01358-f001:**
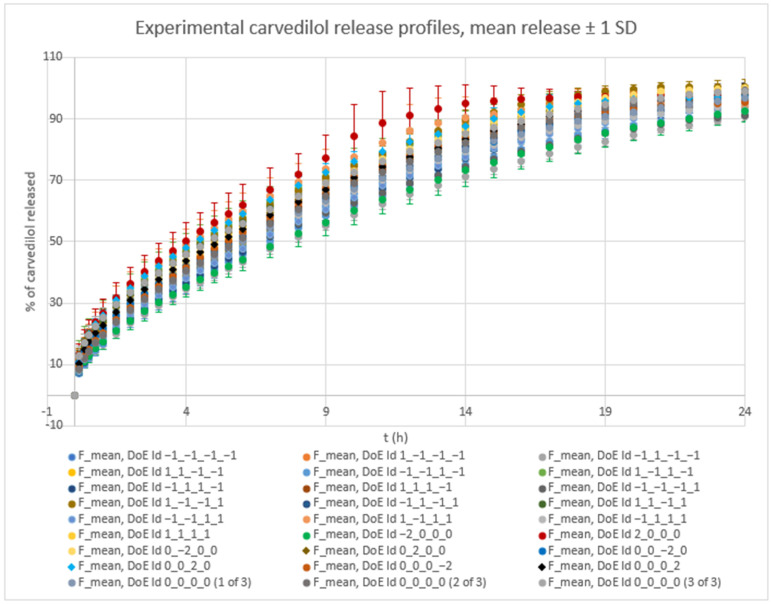
Experimental results of carvedilol release. Mean % carvedilol release ± SD is shown for all performed experiments in the DoE (27 experiments in total). “DoE Id” data in the legend comprised theoretical levels (−2, −1, 0, 1, 2) of factors A (Lac), B (HPMC_Visc), C (HPMC_HP), and D (HPMC_PS) used in each experiment, respectively.

**Figure 2 pharmaceutics-17-01358-f002:**
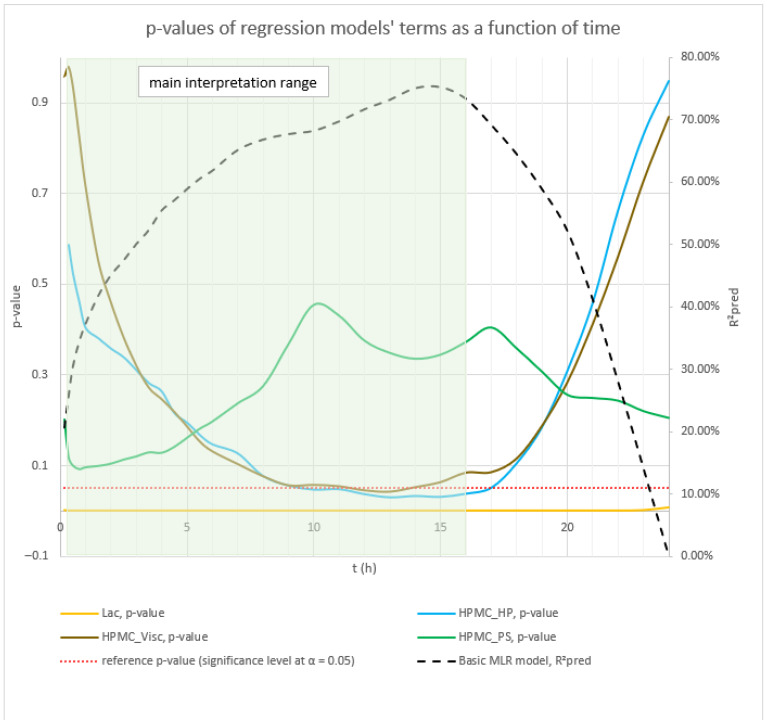
The figure shows the *p*-values of terms in the basic MLR models predicting mean carvedilol release. The dynamic changes in these *p*-values over the release interval reflect shifting term significance, with lower *p*-values indicating stronger effects. Conventionally, *p* ≤ 0.05 (*α* = 0.05) denotes statistical significance. The green background indicates the main interpretation range of the models.

**Figure 3 pharmaceutics-17-01358-f003:**
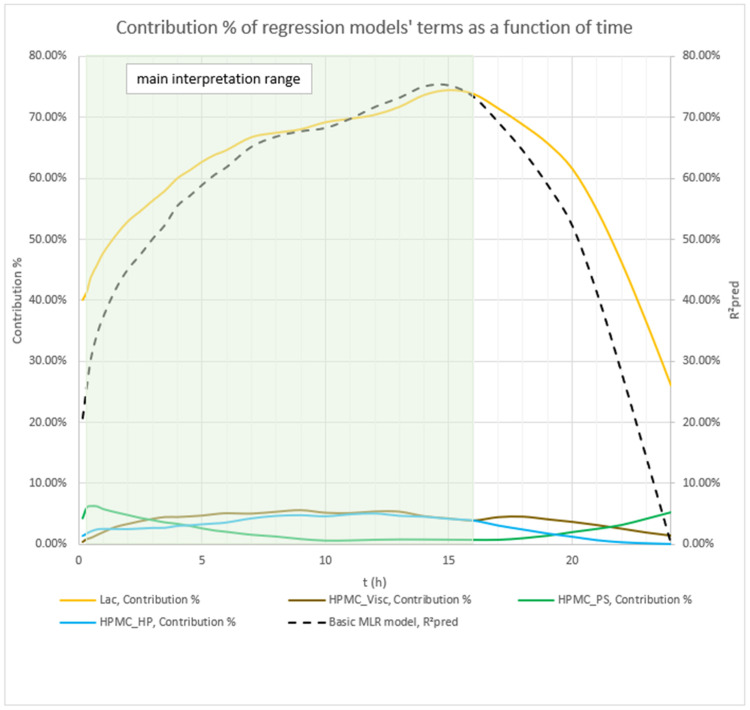
Percentage contribution of terms in the basic MLR models predicting mean carvedilol release. Changes in these contributions over the release interval indicate shifting term importance—higher percentages denote greater influence. The green background indicates the main interpretation range of the models.

**Figure 4 pharmaceutics-17-01358-f004:**
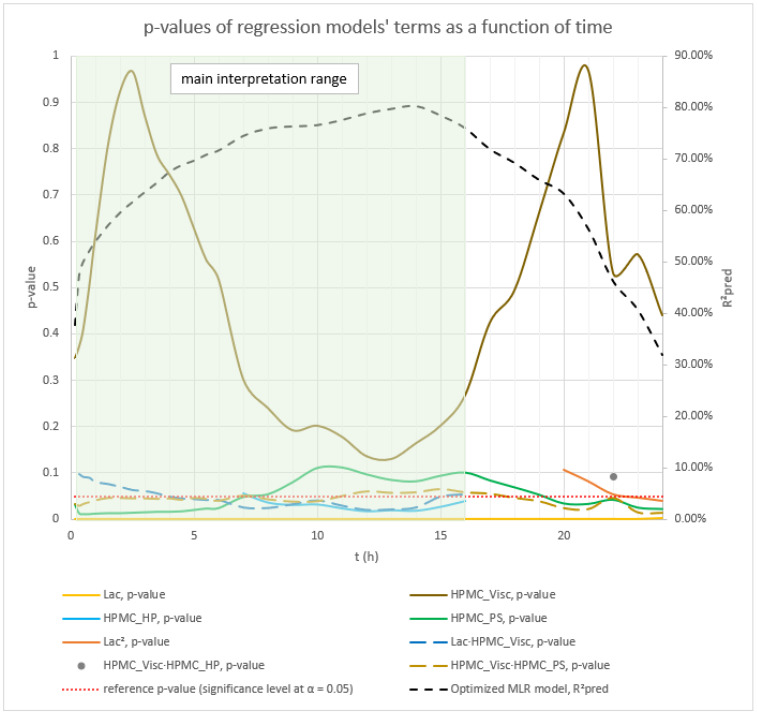
The figure shows the *p*-values of terms in the optimized MLR models predicting mean carvedilol release. The dynamic changes in these *p*-values over the release interval reflect shifting term significance, with lower *p*-values indicating stronger effects. Conventionally, *p* ≤ 0.05 (α = 0.05) denotes statistical significance. The green background indicates the main interpretation range of the models.

**Figure 5 pharmaceutics-17-01358-f005:**
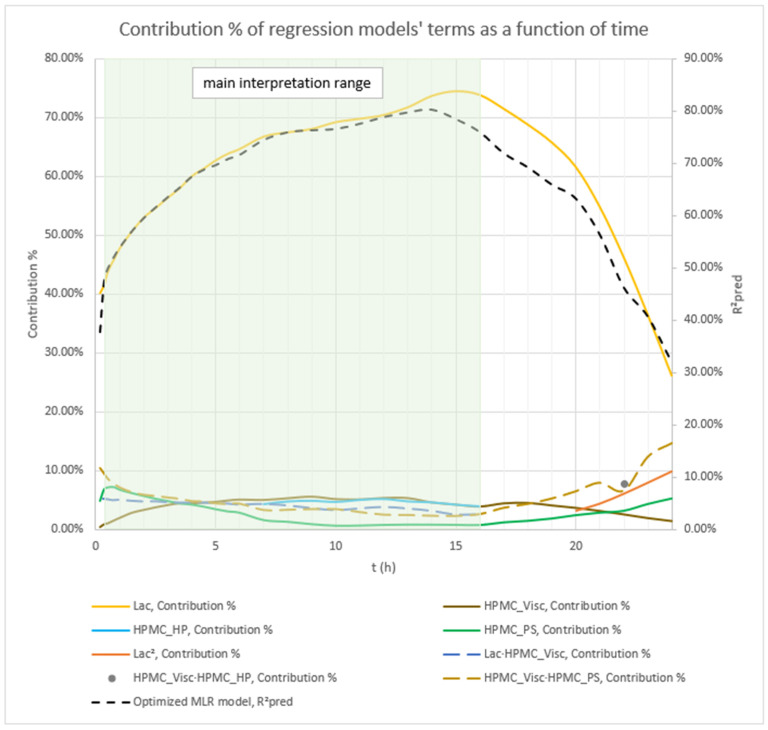
Percentage contribution of terms in the optimized MLR models predicting mean carvedilol release. Changes in these contributions over the release interval indicate shifting term importance—higher percentages denote greater influence. The green background indicates the main interpretation range of the models.

**Figure 6 pharmaceutics-17-01358-f006:**
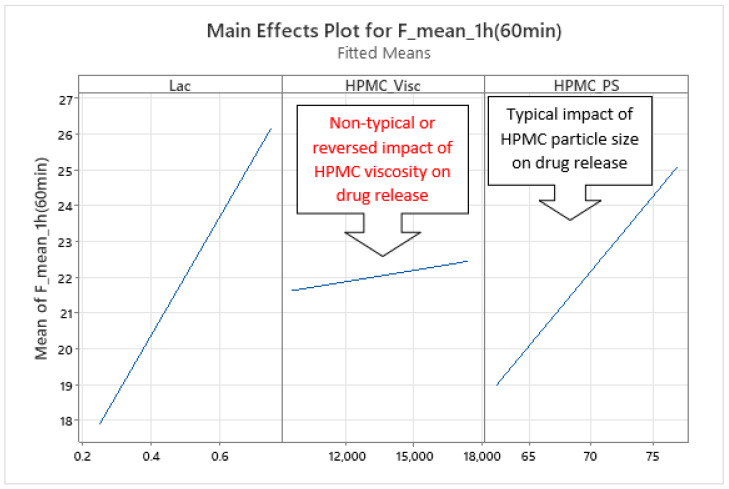
Representative example of main effects influencing mean carvedilol release in Phase 1 of mean release. Data are shown for t = 1 h, though the direction of these effects remained consistent across the timeframe t = 20 min to 2 h (Appendix A). On the *Y*-axis, mean carvedilol release % is presented, and on the *X*-axis, the Lac, HPMC_Visc, and HPMC_PS values, respectively, are presented.

**Figure 7 pharmaceutics-17-01358-f007:**
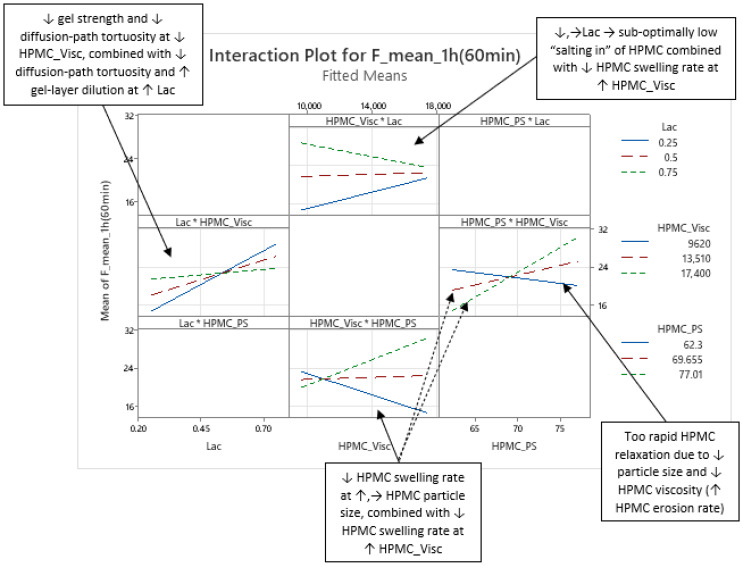
Representative examples of two interactions influencing mean carvedilol release: the interaction between HPMC viscosity (HPMC_Visc) and the lactose fraction in the filler mixture (Lac), and the interaction between HPMC viscosity (HPMC_Visc) and HPMC particle size (HPMC_PS). These interactions are shown here for t = 1 h as an example for Phase 1 of mean release (more examples of these interactions for Phase 1 of mean release are available in Appendix A). However, they are present throughout the entire relevant interpretation range (t = 20 min to t = 16 h), and their nature remains consistent within this period.

**Figure 8 pharmaceutics-17-01358-f008:**
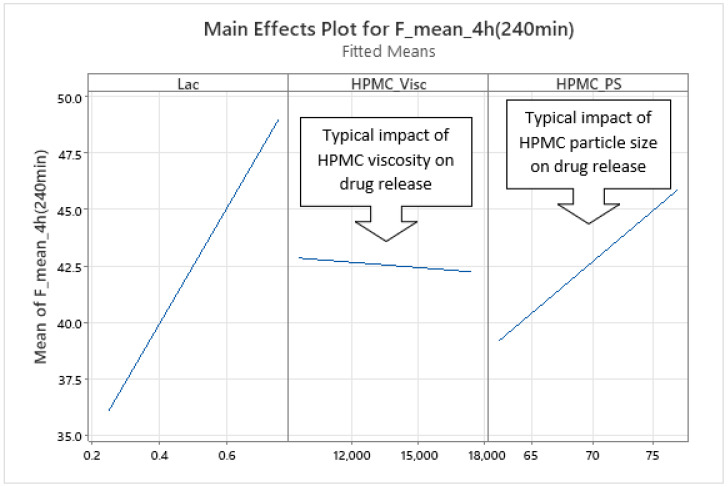
Representative example of main effects influencing mean carvedilol release in Phase 2 of mean release. Data are shown for t = 4 h, though the direction of these effects remained consistent across the timeframe t = 2.5 h to 6 h (Appendix A).

**Figure 9 pharmaceutics-17-01358-f009:**
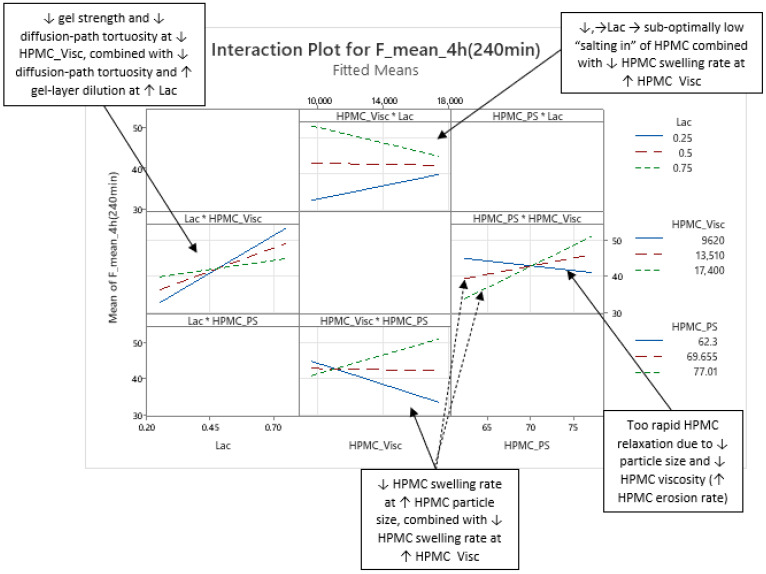
Representative examples of two interactions influencing mean carvedilol release: the interaction between HPMC viscosity (HPMC_Visc) and the lactose fraction in the filler mixture (Lac), and the interaction between HPMC viscosity (HPMC_Visc) and HPMC particle size (HPMC_PS). These interactions are shown here for t = 4 h as an example for Phase 2 of mean release (more examples of these interactions for Phase 2 of mean release are available in Appendix A). However, they are present throughout the entire relevant interpretation range (t = 20 min to t = 16 h), and their nature remains consistent within this period.

**Figure 10 pharmaceutics-17-01358-f010:**
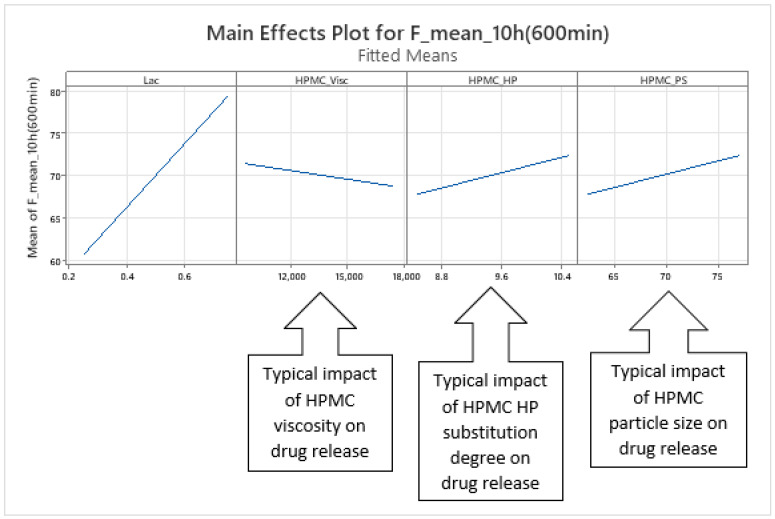
Representative example of main effects influencing mean carvedilol release in Phase 3 of mean release. Data are shown for t = 10 h, though the direction of these effects remained consistent across the timeframe t = 7 h to 16 h (Appendix A).

**Figure 11 pharmaceutics-17-01358-f011:**
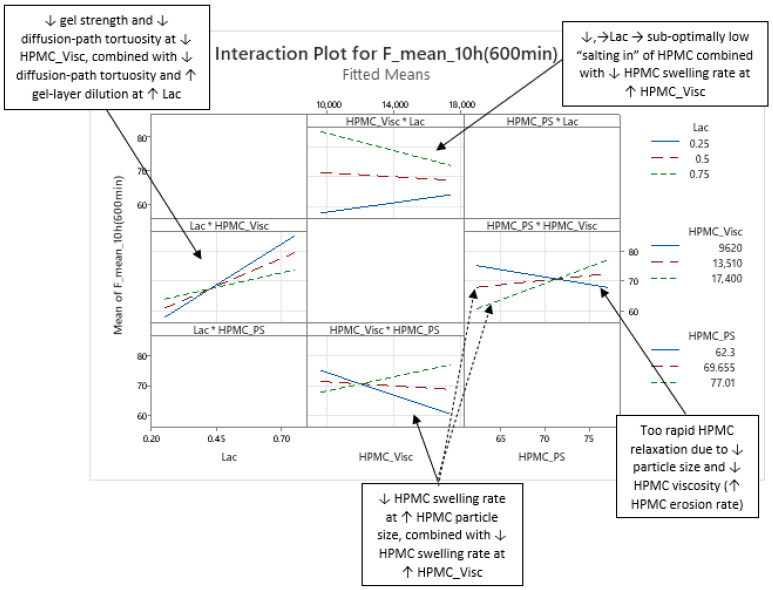
Representative examples of two interactions influencing mean carvedilol release: the interaction between HPMC viscosity (HPMC_Visc) and the lactose fraction in the filler mixture (Lac), and the interaction between HPMC viscosity (HPMC_Visc) and HPMC particle size (HPMC_PS). These interactions are shown here for t = 10 h as an example for Phase 3 of mean release (more examples of these interactions for Phase 3 of mean release are available in Appendix A). However, they are present throughout the entire relevant interpretation range (t = 20 min to t = 16 h), and their nature remains consistent within this period. An asterisk between the names of each pair of factors shown in the figure represents the factors’ second-order in-teraction.

**Figure 12 pharmaceutics-17-01358-f012:**
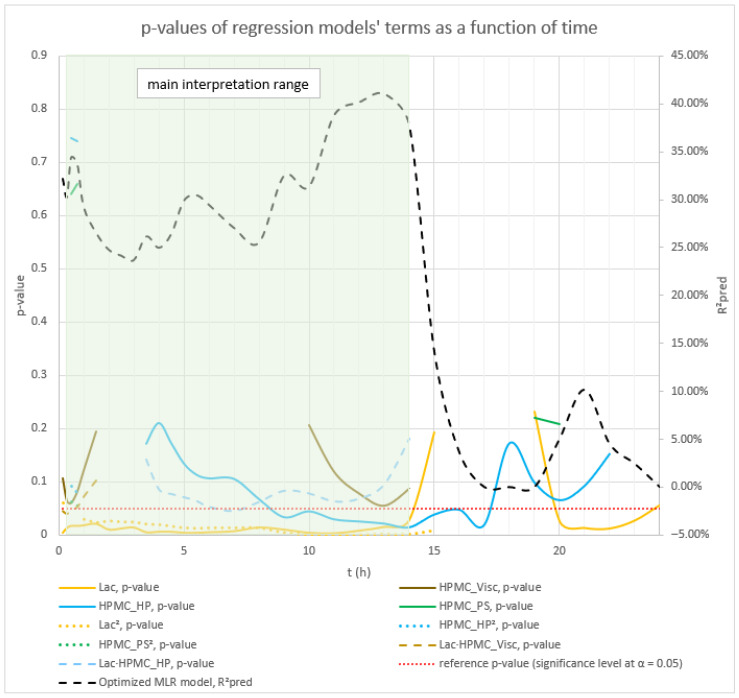
The figure shows the *p*-values of terms in the optimized MLR models predicting SD of carvedilol release. The dynamic changes in these *p*-values over the release interval reflect shifting term significance—with lower *p*-values indicating stronger effects. Conventionally, *p* ≤ 0.05 (α = 0.05) denotes statistical significance. The green background indicates the main interpretation range of the models.

**Figure 13 pharmaceutics-17-01358-f013:**
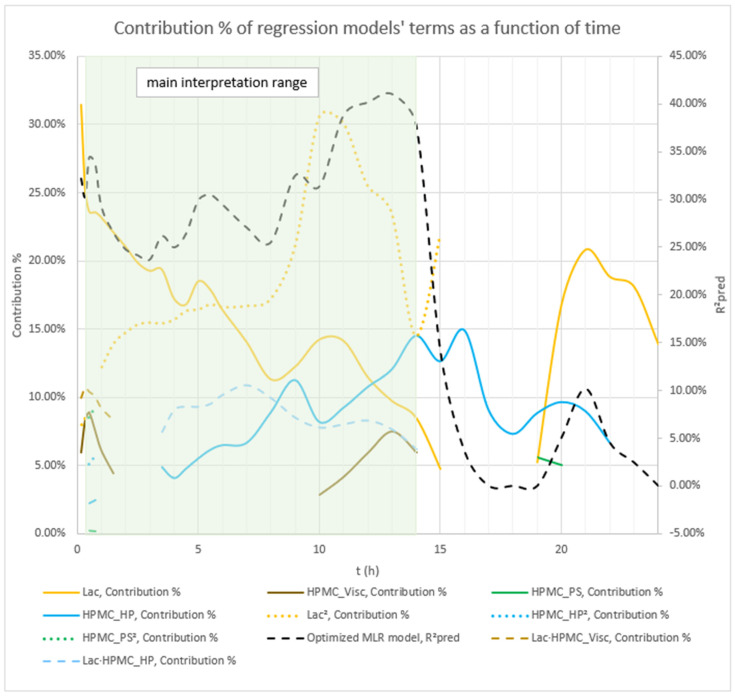
Percentage contribution of terms in the optimized MLR models predicting SD of carvedilol release. Changes in these contributions over the release interval indicate shifting term importance—higher percentages denote greater influence. The green background indicates the main interpretation range of the models.

**Figure 14 pharmaceutics-17-01358-f014:**
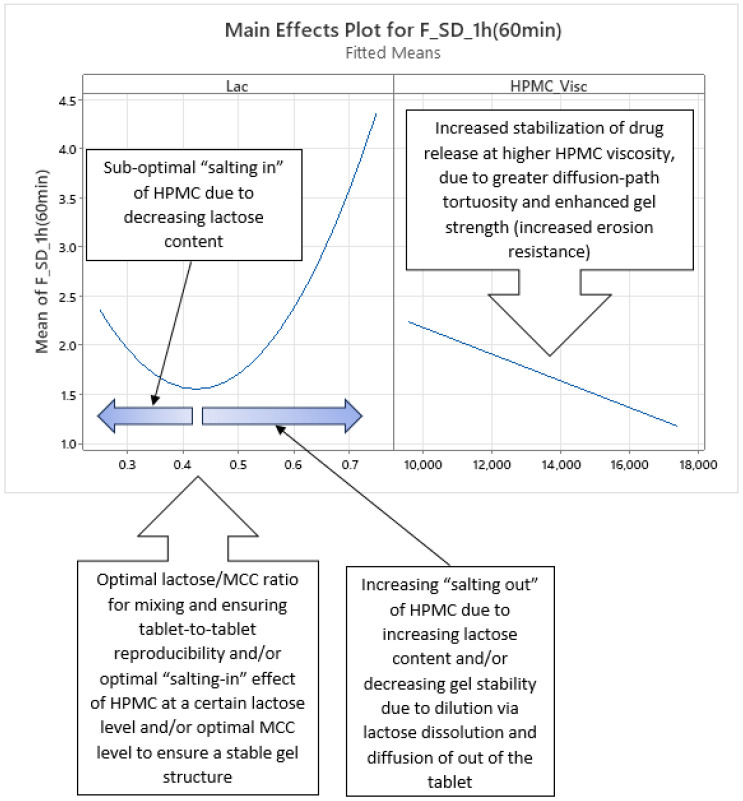
Representative example of main effects influencing SD of carvedilol release in Phase 1 of SD of release. Data are shown for t = 1 h, though the direction of these effects remained consistent across the timeframe t = 20 min to 1.5 h (Appendix A).

**Figure 15 pharmaceutics-17-01358-f015:**
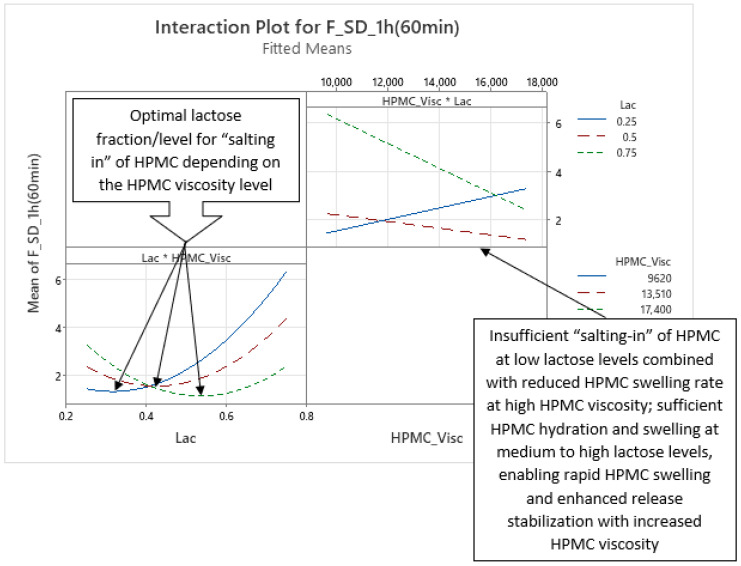
Representative example of an interaction influencing the SD of carvedilol release—the interaction between HPMC viscosity (HPMC_Visc) and the lactose fraction in the filler mixture (Lac). This interaction is shown here for t = 1 h as an example for Phase 1 of SD of release (more examples from Phase 1 are available in Appendix A). An asterisk between the names of each pair of factors shown in the figure represents the factors’ second-order in-teraction.

**Figure 16 pharmaceutics-17-01358-f016:**
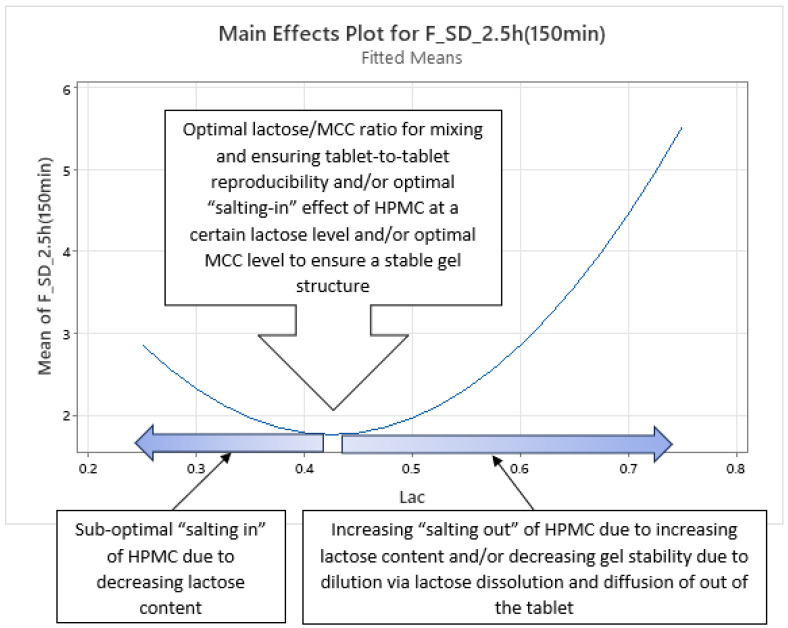
Representative example of a main effect of lactose fraction/level influencing the SD of carvedilol release in Phase 2 of SD of release. Data are shown for t = 2.5 h, though the direction of this effect remained consistent across the timeframe t = 2 h to 3 h (Appendix A).

**Figure 17 pharmaceutics-17-01358-f017:**
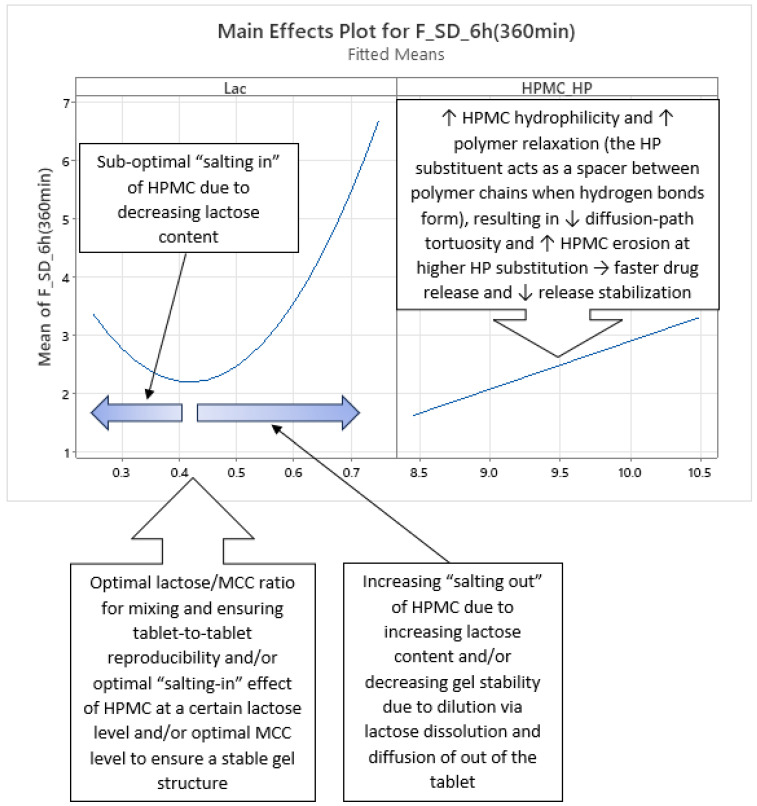
Representative example of main effects influencing SD of carvedilol release in Phase 3 of SD of release. Data are shown for t = 6 h, though the direction of these effects remained consistent across the timeframe t = 3.5 h to 9 h (Appendix A).

**Figure 21 pharmaceutics-17-01358-f021:**
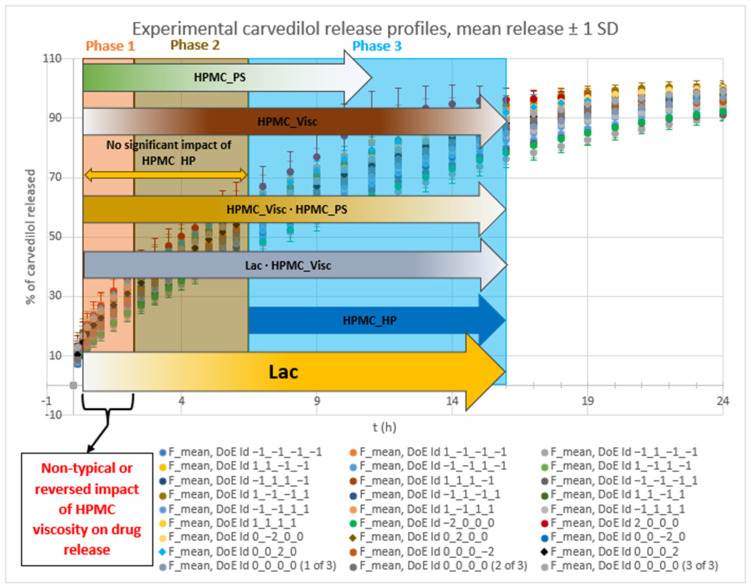
Overview of the main effects and interactions influencing mean carvedilol release across the three phases identified by the optimized MLR models. Fading colours signal effects gradually becoming more important or diminishing with time.

**Figure 22 pharmaceutics-17-01358-f022:**
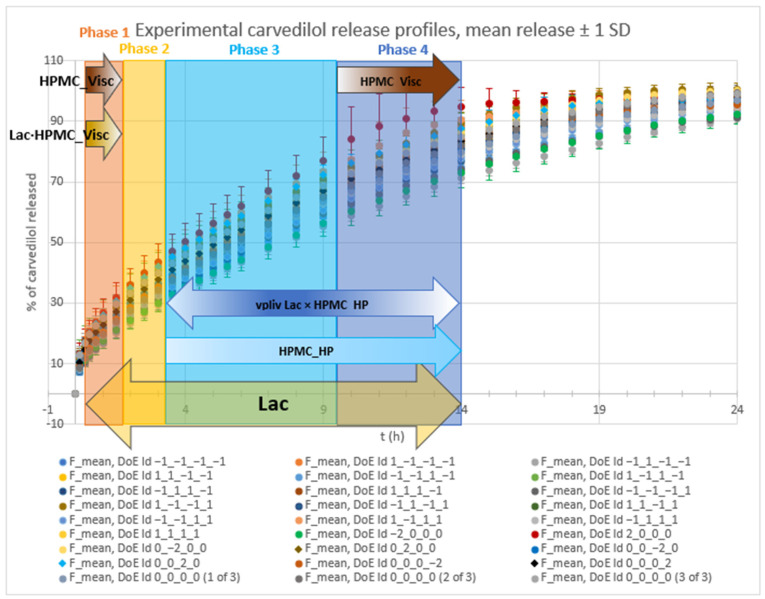
Overview of the main effects and interactions influencing SD of carvedilol release across the four phases identified by the optimized MLR models. Fading colours signal effects gradually becoming more important or diminishing with time.

**Table 1 pharmaceutics-17-01358-t001:** Composition of tablets.

Ingredient	Functionality of Ingredient	The Theoretical Amount of Ingredients in a Tablet (mg)	Theoretical *w*/*w* % of Ingredients in a Tablet	Additional Info
Carvedilol (free base) ^1^ (Ph. Eur.: Carvedilol)	Drug substance ^1^	64.8	10.00	/
METOLOSE^®^ 90SH-15000SR QbD Samples (HPMC 2208 with nominal viscosity 15,000 mPa·s; Ph. Eur.: Hypromellose)	Hydrophilic matrix-forming agent	97.2	15.00	Mixtures of up to 3 QbD samples per experiment were used; see Appendix A for producer analyses data for individual QbD samples in the QbD samples kit
FlowLac^®^ 100 (Ph. Eur.: Lactose Monohydrate)	Water-soluble Filler/Bulking agent/Carvedilol release modifier	Total filler mixture: 475.632 mg; FlowLac^®^ 100 and Avicel^®^ PH-102 each ranging from 118.908 to 356.724 mg in tablets according to the used DoE	Total filler mixture: 73.40 *w*/*w* %; FlowLac^®^ 100 and Avicel^®^ PH-102 each ranging from 18.35 to 55.05 *w*/*w* % in tablets (from 25 to 75 *w*/*w* % in filler mixture) according to the used DoE	/
Avicel^®^ PH-102 (Ph. Eur.: Cellulose, Microcrystalline)	Water-insoluble Filler/Bulking agent/Carvedilol release modifier			/
AEROSIL^®^ 200 Pharma (Colloidal silicon dioxide; Ph. Eur.: Silica, Colloidal Anhydrous)	Glidant	1.944	0.30	/
Magnesium stearate EUR PHAR Vegetable (Ph. Eur.: Magnesium stearate)	Lubricant	8.424	1.30	
Total		648.0	100.0	/

^1^ A model, poorly water-soluble drug, described in PhEur and USP as practically insoluble in water.

## Data Availability

The original contributions presented in this study are included in the article/Appendix A. Further inquiries can be directed to the corresponding author(s).

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
