# Peer review of "Influence of Selected Hypromellose Functionality-Related Characteristics and Soluble/Insoluble Filler Ratio on Carvedilol Release from Matrix Tablets"

_pharmaceutics, 2025, doi:10.3390/pharmaceutics17101358_

Round 1

Reviewer 1 Report

Comments and Suggestions for Authors

This manuscript entitled “Influence of Selected Hypromellose Functionality-related Characteristics and Soluble/Insoluble Filler Ratio on Carvedilol Release from Matrix Tablets” investigates how key functionality-related characteristics (FRCs) of hypromellose (HPMC) – specifically polymer viscosity, degree of hydroxypropoxy substitution, and particle size – as well as the ratio of water-soluble (lactose) to water-insoluble (microcrystalline cellulose, MCC) fillers, affect the release of the poorly water-soluble drug carvedilol from hydrophilic matrix tablets. Overall, the work adds quantitative insight into how HPMC physical-chemical properties and filler composition interplay to shape drug release kinetics, an important practical consideration for generic formulation and QbD development. Specific comments are as follows;

  1. The use of multiple HPMC QbD samples in each formulation to achieve desired FRC levels is novel and pragmatic, but it introduces confounding variability. By blending up to three HPMC batches in one experiment, the actual factor settings (viscosity, substitution, particle size) become averaged values, and intrabatch variability is increased. The authors acknowledge that this approach “reduced the achievable range” of HPMC factors and adds “intrabatch variability” (due to mixing). As a result, it can be challenging to attribute effects unambiguously to individual FRCs versus the mixing strategy. This design limitation should be emphasized more clearly. For example, using physical mixtures assumes linear blending behavior of HPMC properties, which may not hold if polymer interactions occur. Future studies might separately vary each HPMC factor in isolation (e.g. using size fractions or purified grades) to confirm the observed trends without blending confounds.
  2. The dissolution method (900 mL acetate buffer pH 4.5, paddle) is reasonable for the poorly soluble base carvedilol (to maintain sink), and the authors justify it by citing Košir et al. [22]. However, the choice of pH 4.5 (rather than, say, 1.2 or 6.8) could be discussed. Since carvedilol is a weak base, its solubility depends on pH; using gastric pH or bi-phasic media might provide insight to in vivo relevance. The authors should comment on how representative pH 4.5 is of physiological conditions or if it was chosen purely to match their previous work. Additionally, the sampling schedule (starting at 10 min) omits the first 10 minutes due to signal-to-noise considerations. This is acceptable, but it means any very-early “burst” release is unmeasured. Clarifying that the first data point is at 10 min and discussing whether any significant release could have occurred sooner would strengthen the methods description.
  3. The authors use multiple stepwise algorithms (AIC, BIC, α-based) and pick models with highest R²_pred. While stepwise regression is common, it can inflate Type I error (finding spurious terms) and depend heavily on sample noise. Cross-validation or external validation is limited (only 27 experiments). Consideration of alternative methods (e.g. partial least squares or penalized regression) could improve robustness. At minimum, the authors should clarify how they guard against chance correlations given the large number of candidate terms relative to experiments, especially for SD models where data are sparse.
  4. Modeling the SD of release as a response is interesting, but inherently noisy. The authors correctly caution that only four tablets were tested per run, giving limited precision for SD. The R² of SD models was much lower than for mean release. The results (Figures 12–13) are described in detail, but the practical significance of these findings is uncertain. I suggest the authors temper claims about variability effects; with only 4 replicates per point, some identified “significant” interactions (e.g. Lac·HPMC_HP on SD) might be model artifacts. It would help to discuss confidence in the variability models more critically, or to validate key findings (e.g. by replicating selected formulations and checking whether the predicted SD differences hold).
  5. The manuscript emphasizes HP substitution degree (HPMC_HP) as a factor, with some influence on later release and variability phases. However, in formulation practice, HPMC substitution often varies less than viscosity between batches. The actual range of HP values achieved (from Table S1) should be noted when discussing its impact. If the coefficient of variation for HP across mixtures was small, then its observed effect might be subtle or context-specific. A sentence clarifying the magnitude of HP variation (e.g. as given in Table S1 or S3) and its real-world relevance would strengthen interpretation.
  6. The text frequently refers to supplementary Figures (S1–S66) for main effects and interactions. While this is appropriate for detail, key effects should be highlighted in the main figures or a summary plot for clarity. For example, a composite plot or table summarizing which factors are significant at early vs. late times (perhaps a heatmap of p-values) could be included in the main text. This would make the major findings more accessible without diving into the Supplement. As it stands, the reader must trust the textual summaries of S-plots, which can be cumbersome to verify.
  7. The conclusions mostly reflect the data, but two claims deserve scrutiny. The authors state this study is “among the first to control and explore HPMC’s FRC levels systematically in the inter-batch variability context”. This claim appears justified given the specialized design, but it would be good to cite or discuss any closely related prior work explicitly. For instance, the authors cite Košir et al. [22] who also used QbD samples. A brief comparison to that study (e.g. Košir found XYZ with K15M, whereas this work extends to filler effects) would highlight novelty. Additionally, the statement “complete compensation [of variability] is not feasible” could be softened: the data show that adjusting the lactose/MCC ratio only partially mitigates effects, but the conclusion reads as a general principle. Perhaps clarify that within the tested formulation space, 100% compensation wasn’t achieved.
  8. To further strengthen the study, the authors could consider: (a) Expanded replicate experiments for at least a few corner cases (e.g. extreme high-lactose vs. high-MCC) to validate model predictions on mean and variability; (b) Complementary HPMC characterizations, such as gel strength or viscosity measurements of hydrated polymers, to correlate with release outcomes; (c) Evaluation of dissolution at a second pH (e.g. 1.2 or 6.8) to assess pH sensitivity. These suggestions may be out of scope for the present manuscript, but could be noted as future work. Overall, the experimental design is thoughtful, and the systematic analysis is comprehensive. Clarifying some methodological choices and moderating the interpretation of variability findings would improve the manuscript’s rigor.
  9. In Table 1, the layout is hard to parse in text form. In particular, the row for FlowLac®100/MCC shows total filler weight ranges and % ranges, which is useful, but formatting the percentages with ranges in parentheses or in separate columns could improve readability. Check that all table captions clearly explain units (e.g. mg and %w/w).

By addressing these points, the manuscript will be clearer and more polished. Overall, the study is well-conceived and the reporting is thorough; implementing these suggestions will enhance readability and strengthen the rigor of the conclusions.

Author Response

Please find our replies to the reviewer’s comments in the attachment.

Reviewer 2 Report

Comments and Suggestions for Authors

 This manuscript applies a CCD-based, time-resolved analysis to examine how the functional attributes of HPMC and the lactose/MCC ratio influence carvedilol release and within-batch variability. The study is timely and emphasizes a science-based approach to dissolution process design, moving beyond trial-and-error toward rational factor control. However, while informative, several aspects require clarification to improve reproducibility and ensure stronger alignment between methods and intent. I therefore recommend Major Revision.

  1. The description of the USP II dissolution setup would be clearer if key operating parameters were fully specified. Please report the paddle rpm and clarify the ‘flow-through cuvette + autosampler’ pathway (inline vs. offline), including tubing dimensions, hold-up volume, and whether any filtration or optical-clarity control was applied.

  1. Since the dosage form is positioned as an extended/controlled release product, a brief justification for selecting a 24-hour measurement horizon would help align methods with the ER intent. For example, indicate whether 24 h reflects the intended dosing interval or QTPP, confirms complete release or tail-phase kinetics, or addresses late-phase stability.

  1. The dissolution analytics section would benefit from greater methodological detail. Please specify whether withdrawn medium was volume-replaced, on what basis sink conditions were maintained at pH 4.5, and whether samples were filtered prior to UV measurement (including filter type/pore size). If not, please explain how turbidity or particulates were otherwise controlled.

  1. Including a concise summary of tablet physical properties would strengthen the linkage between formulation/process and performance. Reporting hardness, thickness, porosity/density, and friability, along with the associated test methods or instruments, would be helpful.

  1. As all testing was conducted in pH 4.5 medium, please clarify the rationale for this choice. A brief explanation of its biopharmaceutic relevance or solubility constraints, or mention of any spot checks in alternative media/agitation conditions, would help readers interpret the scope of applicability.

Author Response

(The authors gave the same response as above.)

Reviewer 3 Report

Comments and Suggestions for Authors

In this paper the authors investigated how specific functionality-related characteristics of HPMC (viscosity, hydroxypropoxy substitution, particle size) and the ratio of water-soluble to water-insoluble fillers affect carvedilol release from matrix tablets, using a Design of Experiments approach to systematically control HPMC FRC levels.

The abstract could be improved by breaking up long, dense sentences for better readability, slightly condensing the methods description, and emphasizing the practical implications of the findings more clearly.

The study has limited novelty, as the effect of HPMC on controlled drug release has already been extensively investigated, and its contribution lies more in combining previously known factors rather than presenting a truly original discovery, while the claim in the abstract that “this study is among the first to control and explore HPMC’s FRCs systematically” is questionable and may be considered exaggerated.

The use of only four tablets per experiment is a major limitation, insufficient to properly assess intra-batch variability, while the choice of a single dissolution medium (acetate buffer pH 4.5, line 204) reduces clinical relevance since testing at additional pH values that simulate real gastrointestinal conditions is missing, and the blending of up to three types of HPMC QbD samples to reach theoretical levels introduces additional variability and creates confusion in interpreting the results. The statistical strategy of constructing separate MLR models for each time point is questionable, as longitudinal or mixed-effects models would have been more appropriate, while the low R² and R²pred values for SD (25–40% indicate very weak predictive ability, meaning these results should be considered exploratory rather than robust evidence, and the interpretation of “dynamic” p-values is problematic because the authors appear to disregard the fact that multiple testing requires appropriate statistical corrections such as Bonferroni or FDR.

The conclusions drawn from data with high variability (e.g., the standard deviation of release) are forced and uncertain. Only a small portion of the variability is explained by the models, which should be highlighted as a major limitation; however, the text tends to overstate the robustness of the conclusions. Interpretations labeled as “phase 1, 2, 3” (lines 493–523) appear speculative and are not supported by objective validation tests.

The figures are very dense and difficult to follow (e.g., Figures 2–7). The legends are overly long, often repeating information from the main text, and are not sufficiently concise. The supplementary materials (S1–S99) are excessive and hard to manage; a graphical summary or simplified diagram would be far more useful.

Comments on the Quality of English Language

The text is generally understandable, but it contains numerous awkward formulations and repetitions. Examples include:

  • Line 12: “mixtures of up to three QbD samples of HPMC were prepared to achieve target HPMC FRC levels” - redundant and ambiguous phrasing.
  • Lines 202: “dissolution apparatus type 2 with paddles, flow-through cuvettes and an autosampler” - incoherent list; commas are missing before “and.”
  • Lines 213–216: very long sentence, difficult to follow: “Sampling too early… could have resulted… variation would have originated…” - a more concise rewrite is recommended.
  • Lines 374–380 - long sentences, hard to track.
  • Lines 430: “is in accordance with literature mentioned in the introductory section of the article” - overly repetitive; can be simplified to “is consistent with previous literature.”
  • Lines 533–536: “spanning from t = 20 min… to ≈ 2 h (Figures S35–S40). Likely the phase of initial HPMC hydration and swelling, it establishes control over carvedilol release” - fragmented and difficult to read.

English including grammar, style and syntax, should be improved through the professional help from English Editing Company for Scientific Writings.

Author Response

(The authors gave the same response as above.)

Round 2

Reviewer 1 Report

Comments and Suggestions for Authors

The responses are satisfactory.

Reviewer 2 Report

Comments and Suggestions for Authors

I confirm that the issues I raised have been properly addressed, and the manuscript has been significantly improved.

Reviewer 3 Report

Comments and Suggestions for Authors

The authors have significantly addressed the majority of the comments and suggestions provided, and the manuscript has been revised to reflect these changes. Based on the revisions, I believe the manuscript is now suitable for publication in this journal. 

Comments on the Quality of English Language

I recommend that the manuscript undergo a final review by a native English speaker to ensure the language is polished and free from any grammatical or stylistic issues that may affect clarity and readability.